# Graph Structure Inference with BAM:
# Neural Dependency Processing via Bilinear Attention

**Philipp Froehlich**     **Heinz Koeppl**
Department of Electrical Engineering and Information Technology
Technische Universität Darmstadt
{philipp.froehlich, heinz.koeppl}@tu-darmstadt.de

## Abstract

Detecting dependencies among variables is a fundamental task across scientific disciplines. We propose a novel neural network model for graph structure inference, which aims to learn a mapping from observational data to the corresponding underlying dependence structures. The model is trained with variably shaped and coupled simulated input data and requires only a single forward pass through the trained network for inference. Central to our approach is a novel bilinear attention mechanism (BAM) operating on covariance matrices of transformed data while respecting the geometry of the manifold of symmetric positive definite (SPD) matrices. Inspired by graphical lasso methods, our model optimizes over continuous graph representations in the SPD space, where inverse covariance matrices encode conditional independence relations. Empirical evaluations demonstrate the robustness of our method in detecting diverse dependencies, excelling in undirected graph estimation and showing competitive performance in completed partially directed acyclic graph estimation via a novel two-step approach. The trained model effectively detects causal relationships and generalizes well across different functional forms of nonlinear dependencies.

## 1   Introduction

The discovery and understanding of dependencies among variables are fundamental to scientific inquiry across diverse disciplines, such as biology [10], climate science [48], economics [8], and social studies [18]. These dependencies are commonly represented as edges within graphical models, particularly directed acyclic graphs (DAGs), first introduced by Wright [70] to study genetic inheritance, and later advanced for probabilistic inference by Lauritzen and Spiegelhalter [35]. Graph structure inference, the process of deriving such graphical representations from observational data, is essential for gaining insights into complex systems and their underlying causal relationships [49]. A prime example is the estimation of gene regulatory networks from experimental data [19], where identifying dependencies between genes is crucial for understanding biological processes and mechanisms.

Graph structure inference typically employs unsupervised learning methods to estimate the underlying graph through either score-based approaches, which rank graphs by predefined metrics, or constraint-based approaches that identify edges between variable pairs via conditional independence tests [66]. However, these methods face challenges. Score-based approaches encounter computational burdens due to the superexponential growth of potential graph structures with node count, and the necessity to balance fit and structural sparsity [31]. Constraint-based methods usually require a large sample size [66], rely on an elusive optimal threshold hyperparameter , and Shah and Peters [58] proved that the failure of Type I error control in underlying conditional independence tests is unavoidable, which can have significant consequences in downstream analyses.

38th Conference on Neural Information Processing Systems (NeurIPS 2024).

Supervised causal learning techniques, as presented by Lopez-Paz et al. [39, 40, 41], Li et al. [36], Ke et al. [31], Lorch et al. [42], Dai et al. [15], have recently emerged as an appealing alternative to unsupervised methods. In this rising approach, a neural network is typically trained on simulated matrix-shaped data, with corresponding graph structures serving as ground-truth labels for supervised learning. The paradigm capitalizes on the strengths of deep learning to discern complex patterns in data, thereby enabling accurate graph structure inference.

Supervised causal learning strongly relies on neural networks' ability to extract dependency information from observational data matrices. Current approaches for processing observational data, whether in causal learning [31, 42], or related fields [34, 54], typically embed the data into expanded observational data spaces, attempting to implicitly capture dependencies within the Euclidean space in transformed data matrices. After marginalizing over the sample dimension, existing supervised causal learning methods [31, 42] then predict edges separately from each other via compressed observational representation vectors, rather than estimating a single graph object as in graphical lasso (GLASSO) algorithms [72, 7, 17]. GLASSO methods effectively optimize inverse covariance matrices in SPD space, where matrix entries directly encode the graph's conditional dependencies. This classical, highly influential approach is particularly powerful in the Gaussian case, where the covariance matrix is a sufficient statistic for the distribution. We extend this geometric insight from classical optimization to the neural network era by introducing SPD layers that learn and process dependency information directly. Instead of using pre-derived mathematical operations as in GLASSO-based algorithms, our network learns these shape-invariant transformations from data. Combined with learned transformations in the observational space, our approach generalizes beyond the Gaussian assumption of classical methods while preserving both the theoretical guarantees and computational efficiency of GLASSO approaches. In contrast to GLASSO methods which must solve an optimization problem for each new dataset, our approach requires only a single forward pass.

Analogous to GLASSO-based methods, the symmetry of covariance matrices constrains our approach to undirected graph estimation. We extend beyond this limitation via a novel two-phase procedure. We first estimate the graph's skeleton and immoralities, then identify edge directions to obtain a completed partially directed acyclic graph (CPDAG) [12]. While this follows the general principle of the PC-algorithm [61], our deep learning approach distinguishes between skeleton edges and moralized edges through joint optimization in SPD space, instead of relying on sequential conditional independence tests over all possible immoralities for edge identification. By efficiently identifying immoralities in the first step, our method drastically reduces the number of required independence tests, minimizing both computational complexity and the propagation of false positive orientation errors.

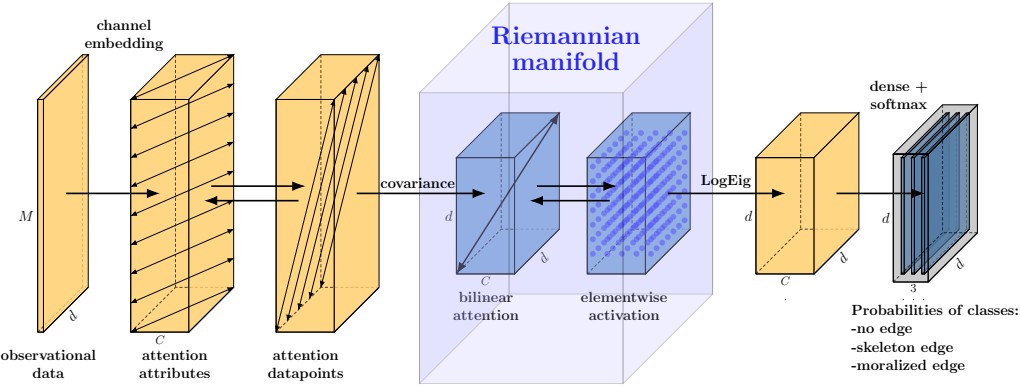

Figure 1: Neural network architecture: An input of arbitrarily shape $(M, d)$ is provided, which is then embedded into $C$ channels. Attention between attributes and attention between datapoints are applied alternately. Covariance matrices are calculated, followed by alternating applications of bilinear attention and the custom activation function in the Riemannian manifold of SPD matrices. The matrices are then transformed back into Euclidean space using the Log-Eig layer. Output probabilities for each pair of variables being in the classes "no edge", "skeleton edge", and "moralized edge" are calculated using dense layers and applying a softmax layer on the channel axis.

## 2 Method

### 2.1 Supervised approach for learning graph structures

In supervised graph learning, simulated training data is generated in the form of input/label pairs $(\boldsymbol{X}, \mathcal{G})$. Each pair consists of an observational data matrix $\boldsymbol{X} \in \mathbb{R}^{M \times d}$, where $M$ denotes the number of samples and $d$ represents the number of attributes, and its corresponding graph structure $\mathcal{G} = (V, E)$. The graph $\mathcal{G}$ is defined by a set of nodes $V = \{v_1, \ldots, v_d\}$ and a set of edges $E \subseteq V \times V$. Alternatively, graphs can be represented as adjacency matrices $\boldsymbol{A} \in \{0, 1\}^{d \times d}$, where $\boldsymbol{A}_{ij} = 1$ if an edge exists between nodes $v_i$ and $v_j$, and $\boldsymbol{A}_{ij} = 0$ otherwise. The process begins by creating a random graph, followed by generating observational data matrices using a structural equation model (SEM) $X_v = f_v(X_{\mathrm{pa}_{\mathcal{G}}(v)}, \epsilon_v)$, where $f_v$ is a measurable coupling function of parent nodes $\mathrm{pa}(\mathcal{G})$ and zero-mean error $\epsilon_v$. The random coupling in the SEM is designed to approximate a broad class of functions, enabling the model to perform classification of the presence or absence of causal relationships, irrespective of their specific functional forms. A neural network then learns the relationship between the graph and the generated data matrix as a mapping $\boldsymbol{X} \mapsto \mathcal{G}$, enabling the inference of graph structures from observational data. The supervised causal learning procedure is shown in Figure 2.

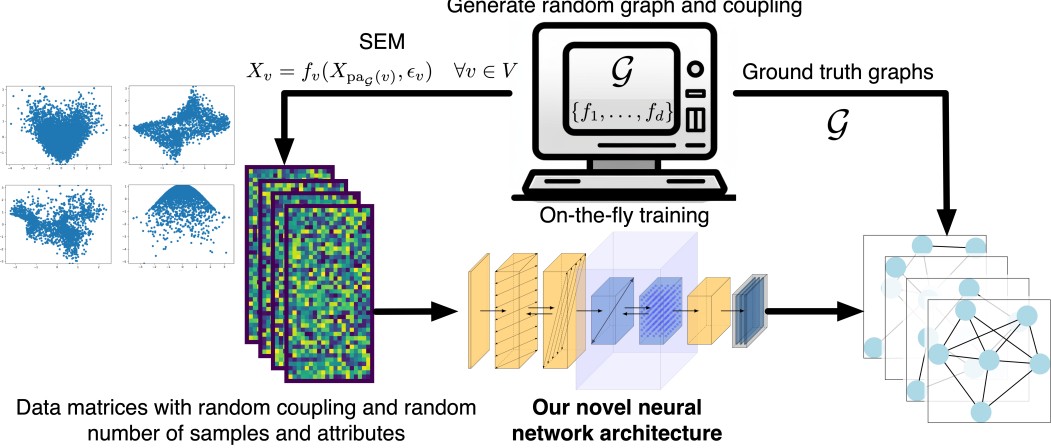

Figure 2: Overview of our supervised graph learning approach. We generate synthetic training data by first creating random ground truth graphs. Then, observational data matrices are generated using SEMs with randomly parameterized polynomials. Employing our novel neural network architecture, we learn the mapping $\boldsymbol{X} \mapsto \mathcal{G}$ from observational data matrices to the corresponding graph structures.

### 2.2 Generation of training data

For training data generation, we utilize Erdős–Rényi (ER) graphs, denoted as $ER(d, q)$, where the number of nodes $d$ and the expected degree $q$ are sampled from discrete uniform distributions $d \sim \mathcal{U}(\{\underline{d}, \ldots, \overline{d}\})$ and $q \sim U([\underline{q}, \overline{q}])$, respectively. The shape-invariant design of our model enables it to train without restriction to a fixed $(d, q)$ pair, which proves advantageous when the graph density is unknown. For each graph $\mathcal{G}_i$, we generate a data matrix $\boldsymbol{X}_i \in \mathbb{R}^{M \times d}$ using an SEM, where the sample size $M$ is drawn from a discrete uniform distribution, $M \sim \mathcal{U}(\{\underline{M}, \ldots, \overline{M}\})$. Although ER graphs are used for training, we demonstrate that our approach maintains its performance when tested on graphs generated using different random graph models, as detailed in Appendix G.6.

As coupling functions in the SEM, we employ random multivariate Chebyshev polynomial functions to generate diverse continuous training data. Chebyshev polynomials effectively approximate real-world functions, showing factorial decay in their coefficients given that higher order derivatives are bounded [63], i.e., for the $n$-th coefficient $c_n$ it holds $|c_n| \leq \frac{C}{n!}$ for a constant $C$, making higher-degree terms negligible, such that all such smooth functions can be efficiently approximated with a

low polynomial degree Chebyshev polynomial. This choice of parameterization avoids unnatural, non-smooth coupling functions where a higher-order derivative is much larger than a lower-order one, and where there is a hard cutoff in the approximated order of the derivative. Chebyshev polynomials have been widely used in various applications due to their excellent approximation properties and computational efficiency [44, 53]. Scatterplots of input/output values of four example Chebyshev polynomials are shown on the left side of Figure 2. The error terms $\epsilon_v$ in the SEM are modeled using Gaussian mixture models, which provide a flexible and expressive framework for capturing a diverse range of error distributions [55]. Details about the parameterization of the SEM can be found in Appendix E.

## 2.3 Prediction task: three-class edge classification

Our neural network architecture is designed to process SPD matrices, which inherently contain exclusively symmetrical relationship information, thus excluding non-symmetrical information necessary for estimating edge directionality. While this configuration prevents the estimation of a DAG or CPDAG in a single run, it enables efficient estimation of symmetrical structures such as the graph skeleton. Additionally, it facilitates the identification of moralized edges. We differentiate between the following edge types:

- **Skeleton edges**: Undirected edges found in the underlying DAG.
- **Moralized edges**: Not present in the DAG but emerge due to conditional dependencies among nodes sharing a common child without a connecting edge between the parents.
- **No edge**: Conditionally independent variables given all other variables.

Skeleton and moralized edges are both represented as undirected edges in models that use partial correlations or conditional independence tests, in which the conditioning set includes all other variables, to estimate the moral graph, as e.g., in [17, 59].

Using the assumption of faithfulness and the Markov condition enable us to identify the Markov equivalence class [61] and thus uniquely solve the three-class classification problem. Appendix C provides an explicit proof outlining the conditional independence relations that lead to a unique solution.

The prediction targets are an extension of the binary adjacency matrix $\boldsymbol{A} \in \{0, 1\}^{d \times d}$ to a set of one-hot encoded adjacency matrices, denoted as $\widetilde{\boldsymbol{A}} \in \{0, 1\}^{d \times d \times 3}$. Here, for each $i, j$, the vector $\widetilde{\boldsymbol{A}}_{i,j,\cdot} \in \{0, 1\}^3$ represents a one-hot encoded classification among the three classes: *skeleton edge*, *moralized edge*, and *no edge*.

## 2.4 Neural network architecture

We use the multi-dimensional analogue of matrix multiplication: For a tensor $\boldsymbol{A} \in \mathbb{R}^{I \times J \times K}$ and a matrix $\boldsymbol{B} \in \mathbb{R}^{K \times L}$ we denote

$$\boldsymbol{AB} = \boldsymbol{C} \in \mathbb{R}^{I \times J \times L} \quad \text{with} \quad \boldsymbol{C}_{ijl} = \sum_k \boldsymbol{A}_{ijk} \boldsymbol{B}_{kl}. \tag{1}$$

The network architecture is depicted in Figure 1. In the following, we introduce the individual layers.

**Channel embedding.** We perform an embedding of the input $\boldsymbol{X} \in \mathbb{R}^{M \times d}$ to obtain a hidden representation with $C$ channels. For this, one axis for $\boldsymbol{X}$ is extended to $\widetilde{\boldsymbol{X}} \in \mathbb{R}^{M \times d \times 1}$ and then trainable weights $\boldsymbol{W}_1 \in \mathbb{R}^{1 \times C}$, $\boldsymbol{W}_2 \in \mathbb{R}^{C \times C}$ are used to obtain

$$\boldsymbol{H} = \widetilde{\boldsymbol{X}} + \text{relu}\left(\widetilde{\boldsymbol{X}} \boldsymbol{W}_1\right) \boldsymbol{W}_2 \in \mathbb{R}^{M \times d \times C},$$

where broadcasting is used for the addition.

A shape- and permutation-invariant neural network is achieved by employing $C \times \widetilde{C}$ weight matrices in conjunction with attention mechanisms for fixed channel dimensions $C$ and $\widetilde{C}$ of the current and the next layer. This enables information flow among elements both within each matrix and across the $C$ matrices, analogous to sequence models like the Transformer [65], where a sequence of length $l$ is embedded into an $l \times C$ matrix and processed via self-attention with $C \times \widetilde{C}$ trainable weights.

**Observational data self-attention.** Self-attention for $C$ observational $M \times d$ data matrices, stored in a tensor $\boldsymbol{X} \in \mathbb{R}^{M \times d \times C}$, proposed by Kossen et al. [34], is based on axial attention [26, 67] and commonly used for processing observational data [4, 31, 42]. The attention-between-attributes layer (Figure 3, left) computes $M$ attention matrices $\boldsymbol{A}$ of shape $d \times d$, allowing attributes to attend to each other in parallel for each sample. The attention-between-samples layer interchanges $M$ and $d$, resulting in $d$ attention matrices of shape $M \times M$, enabling samples to attend to each other. These layers capture information flow along both axes of the observational data matrices. A mathematical description is provided in Appendix B.1.

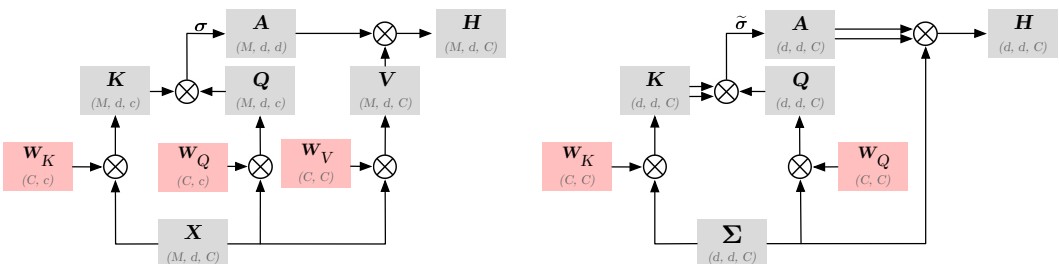

Figure 3: Left: Observational data self-attention layer across attributes. Gray denotes non-trainable tensors, and red represents trainable weights. Matrix multiplication is performed after necessary transposition to match axis dimensions. Right: Bilinear self-attention layer. The double arrow signifies the use of the matrix as a bilinear operator.

**Transition to SPD space via covariance calculation.** Let a transformed data matrix $\boldsymbol{X} \in \mathbb{R}^{M \times d \times C}$ be given. To obtain explicit dependency information, we calculate channelwise covariance matrices, i.e.:

$$\boldsymbol{\Sigma} = \frac{1}{M-1} \left( (\boldsymbol{X} - \boldsymbol{\mu_X})^{T(3,2,1)} \odot (\boldsymbol{X} - \boldsymbol{\mu_X})^{T(3,1,2)} \right)^{T(2,3,1)} \in \mathbb{R}^{d \times d \times C},$$

where we define for tensors $\boldsymbol{A} \in \mathbb{R}^{I \times J \times K}$, $\boldsymbol{B} \in \mathbb{R}^{I \times K \times L}$ the $I$-parallel matrix multiplication $\boldsymbol{A} \odot \boldsymbol{B} := (\boldsymbol{A}_{i,\cdot,\cdot} \boldsymbol{B}_{i,\cdot,\cdot})_{i=1,\ldots,I} \in \mathbb{R}^{I \times J \times L}$, and $\boldsymbol{\mu_X} = \frac{1}{M} \mathbf{1}_M^T \boldsymbol{X} \in \mathbb{R}^{1 \times d \times C}$ is a tensor of sample means. We denote by $\mathcal{S}_{\succeq}^{d \times d}$ the cone of $d \times d$ symmetric positive semi-definite (SPD) matrices, and by $\mathcal{S}_{\succeq}^{d \times d \times C} = \mathcal{S}_{\succeq}^{d \times d} \times \cdots \times \mathcal{S}_{\succeq}^{d \times d}$ the $C$-ary Cartesian power of $\mathcal{S}_{\succeq}^{d \times d}$. It holds $\boldsymbol{\Sigma} \in \mathcal{S}_{\succeq}^{d \times d \times C}$.

The transpose notation $^{T(i,j,k)}$ permutes the dimensions of a tensor according to the specified order $(i, j, k)$, following the commonly used notation in deep learning [1]. For example, $\boldsymbol{A}^{T(2,3,1)}$ rearranges the dimensions of $\boldsymbol{A}$ such that the second dimension becomes the first, the third becomes the second, and the first becomes the third.

**Introducing the bilinear attention mechanism.** To effectively process dependency information stored in the covariance matrices, which is crucial for causal discovery, we propose a novel bilinear attention mechanism. Inspired by the success of SPD networks in image processing, our bilinear attention layer is designed to be shape-invariant and permutation-invariant. In contrast to existing SPD architectures [28, 68] that typically parameterize weights $\mathbf{W} \in \mathbb{R}^{d \times d_{\text{out}}}$ to be applied as $\mathbf{W}^T \boldsymbol{\Sigma} \mathbf{W} \in \mathcal{S}_{\succeq}^{d \times d}$ to a matrix $\boldsymbol{\Sigma} \in \mathcal{S}_{\succeq}^{d \times d}$, we parameterize weights $\boldsymbol{W} \in \mathbb{R}^{C \times C}$ to act as linear combinations $\boldsymbol{\Sigma} \boldsymbol{W} \in \mathcal{S}_{\succeq}^{d \times d \times C}$ on a set of covariance matrices $\boldsymbol{\Sigma} \in \mathcal{S}_{\succeq}^{d \times d \times C}$. We employ an attention mechanism to generate attention matrices $\mathbf{A} \in \mathbb{R}^{d \times d \times C}$, serving as adaptable $d \times d$ weights on each $d \times d$ matrix of $\boldsymbol{\Sigma} \in \mathbb{R}^{d \times d \times C}$. Our bilinear attention mechansim is shown in (Figure 3, right).

The Riemannian manifold of SPD matrices has nonpositive sectional curvature, and its geometric properties are not preserved under Euclidean operations [9], leading to issues when using traditional neural networks [46]. Specialized architectures respecting the SPD manifold's geometry are necessary. Bilinear[1] matrix multiplication $\boldsymbol{\Sigma} \to \boldsymbol{W}^T \boldsymbol{\Sigma} \boldsymbol{W}$, analogous to a dense layer in Euclidean space, preserves the space of SPD matrices $\mathcal{S}_{\succeq}^{d \times d}$ and serves as a primary tool for SPD layers [68].

---

[1]The mapping $\boldsymbol{A} \mapsto \boldsymbol{A} \boldsymbol{\Sigma} \boldsymbol{A}^T$ is quadratic and hence nonlinear in $\boldsymbol{A}$. It is often referred to as bilinear as it is a special case of the mapping $(\boldsymbol{A}, \boldsymbol{B}) \mapsto \boldsymbol{A} \boldsymbol{\Sigma} \boldsymbol{B}^T$.

For an input $\boldsymbol{\Sigma} \in \mathcal{S}_{\succeq}^{d \times d \times C}$, we obtain keys $\boldsymbol{K} = \boldsymbol{\Sigma} \boldsymbol{W}_K \in \mathcal{S}_{\succeq}^{d \times d \times C}$ and queries $\boldsymbol{Q} = \boldsymbol{\Sigma} \boldsymbol{W}_Q \in \mathcal{S}_{\succeq}^{d \times d \times C}$, using non-negativity constraints on the weights $\boldsymbol{W}_K, \boldsymbol{W}_Q \in \mathbb{R}_+^{C \times C}$.

Keys and queries are now combined in a bilinear fashion, parallel over the $C$ channels by calculating
$$\boldsymbol{K} \otimes \boldsymbol{Q} := \big(\boldsymbol{K}_{\cdot,\cdot,c} \boldsymbol{Q}_{\cdot,\cdot,c} \boldsymbol{K}_{\cdot,\cdot,c}\big)_{c=1,\dots,C} \in \mathcal{S}_{\succeq}^{d \times d \times C}, \tag{2}$$
resulting in a tensor of $C$ SPD matrices.

We replace the softmax function for traditional attention with a custom softmax function $\widetilde{\boldsymbol{\sigma}}$ to preserve positive definiteness, i.e.,
$$\widetilde{\boldsymbol{\sigma}} : \mathcal{S}_{\succeq}^{d \times d} \to \mathcal{S}_{\succeq}^{d \times d} \qquad \widetilde{\boldsymbol{\sigma}}(\boldsymbol{S}) := \sqrt{\boldsymbol{\Lambda}(\boldsymbol{S})} \exp[\boldsymbol{S}] \sqrt{\boldsymbol{\Lambda}(\boldsymbol{S})} \quad \text{where} \quad \boldsymbol{\Lambda}(\boldsymbol{S}) := \operatorname{diag}\left(\frac{1}{\exp[\boldsymbol{S}]\mathbf{1}_d}\right), \tag{3}$$
where $\exp[\cdot]$ denotes the elementwise application of the exponential function, $\mathbf{1}_d$ is a vector of length $d$ with all entries being 1, and $\operatorname{diag}$ transforms a vector of length $d$ into a $d \times d$ diagonal matrix. The quotient and root are also taken elementwise.

Using the definition of the bilinear parallel tensor product in 2,we obtain the attention matrix $\boldsymbol{A}$ by applying $\widetilde{\boldsymbol{\sigma}}$ channelwise, i.e.,
$$\boldsymbol{A} := \Big(\widetilde{\boldsymbol{\sigma}}((\boldsymbol{K} \otimes \boldsymbol{Q})_{\cdot,\cdot,c})\Big)_{c=1,\dots,C} \in \mathcal{S}_{\succeq}^{d \times d \times C},$$
and we obtain the output of the bilinear layer as
$$\boldsymbol{H} = \boldsymbol{A} \otimes \boldsymbol{\Sigma} \in \mathcal{S}_{\succeq}^{d \times d \times C}.$$

The elementwise exponential function preserves positive definiteness because SPD matrices are closed under addition and the Hadamard product [9], and $\exp[\boldsymbol{S}] = \sum_{n=0}^{\infty} \frac{1}{n!}[\boldsymbol{S}]^n$, with elementwise exponentiation $[\cdot]^n$. Theorems regarding the eigenvalue regularization and stability properties of our modified softmax function $\widetilde{\boldsymbol{\sigma}}$ can be found in Appendix A.

**Log-Eig layer and output softmax.** Data representation transitions from the SPD space to Euclidean space through the Log-Eig layer, as proposed by [28]. This transformation, given an input matrix $\boldsymbol{S} = \boldsymbol{U}\boldsymbol{D}\boldsymbol{U}^T$ via eigendecomposition can be expressed as
$$l : \mathcal{S}_{\succeq}^{d \times d} \to \mathbb{R}^{d \times d}, \quad l(\boldsymbol{S}) := \log(\boldsymbol{S}) := \boldsymbol{U}\log(\boldsymbol{D})\boldsymbol{U}^T.$$
The final layer, equipped with a softmax activation function, consists of $C = 3$ output units for generating the probabilities.

**Interpretation.** In standard attention, the $(i,j)$-th entry of the attention matrix $\boldsymbol{A}$ directly represents a scalar influence of element $i$ on element $j$. In our bilinear attention framework, however, interdependencies emerge. Specifically, for an output pair $(i,j)$, the corresponding output value is given by the bilinear form $\sum_{k,l} A_{i,k} S_{k,l} A_{l,j}$. Hence, the influence on the $(i,j)$-th entry depends not only on $A_{i,j}$ but on the entire $i$-th row and $j$-th column of $\boldsymbol{A}$. This produces a cross-shaped attention pattern within $\boldsymbol{A}$ and allows columns of $\mathbf{S}$ to attend to each other, capturing more complex interactions. The learned SPD representations can be interpreted as end-to-end learned kernel matrices encoding different kinds of dependencies across multiple channels. A detailed discussion of the intuition behind our bilinear attention mechanism is provided in Appendix I.

**Implementation details.** Additional details regarding the activation function, residual connections and normalization, the use of multiple heads in attention layers and the initialization of the $C \times C$ weights in the BAM layers are provided in Appendices B.2, B.3, B.4, B.5, respectively.

**CPDAG estimation from the graph skeleton and the set of moralized edges.** To derive the CPDAG from the graph skeleton and identified immoralities, we train a second neural network to infer v-structures from two parent nodes together with potential common child nodes that have edges to both parents, as well as other neighbor nodes related to these parents. We iterate over all inferred immoralities, treating the two nodes involved in each immorality as parent nodes. By applying distinct layers to the data corresponding to the parent nodes, potential common children, and neighbors, we can break the symmetry among nodes and enable role-specific learning, thus facilitating edge directionality inference. The resulting CPDAG is further refined using Meek rules [45]. Details about the CPDAG estimation step can be found in Appendix D.

# 3 Related Work

Causal discovery primarily relies on unsupervised learning methods, including constraint-based approaches that infer conditional independencies [61, 16] and score-based methods that optimize a score function under acyclicity constraints [13, 71]. Some studies have estimated relations from multi-sequence alignment (MSA) data matrices, such as Rao et al. [54] using axial attention [26] and Li et al. [37] employing a convolutional neural network on precision matrices. However, these models are designed for the specific conditions of the MSA prediction task and its fixed data dimension of 20 naturally-occurring residue types.

Our work closely aligns with Ke et al. [31] and Lorch et al. [42], which also use attention mechanisms between samples and attributes to estimate adjacency matrices. The methodologies differ in their strategies for deriving an adjacency matrix from observational data representations. [31] employ an autoregressive transformer approach, whereas [42] utilize the dot product of embedding vectors derived from max-pooling across the sample axis of the observational data matrices. We demonstrate that deriving output adjacency matrices can be efficiently achieved through covariance matrices, allowing for direct processing of dependency information and enhanced learning efficiency through geometric learning on the SPD manifold.

Unlike [31], our model can be trained across different shapes of sample and attribute numbers, eliminating the need for re-training on different datasets. While [42]'s model can evaluate data with varying dimensions, their implementation has practical limitations in training across variable sample sizes, and it is restricted to handling a limited range of dimensions for training. Our model addresses identifiability challenges by first estimating an undirected graph, then proceeding to CPDAG estimation by testing immoralities and using Meek's rules. In contrast, [31] and [42] deduce a DAG using both observational and interventional data such that these models dependent on the availability of interventional data or making random guesses for edges that cannot be directly inferred.

Closely related to our approach, Li et al. [36] used permutation-invariant models for supervised causal inference, but their method directly computes correlation matrices without learning a high-dimensional representation of the observational data, which may limit its efficacy when covariance matrices are not sufficient. Despite preserving invariances, the small number of 5 free parameters per layer could restrict its representational power.

[39, 40] proposed a supervised learning framework using kernel mean embeddings. [43] address identifiability with independence tests and cascade classifiers trained on vicinal graphs specific to observational data. Our method adopts a more general approach, training a network that does not require re-training for new evaluation samples. [15]'s work on immoralities aligns with our second CPDAG estimation step.

# 4 Experiments

**Training data hyperparameter settings.** The hyperparameters for sampling training data, which are defined in section 2.2, are set to $\underline{d} = 10$, $\overline{d} = 100$, $\underline{q} = 1$, $\overline{q} = \min(\frac{d}{3}, 5)$, $\underline{M} = 50$, and $\overline{M} = 1000$. This configuration allows for denser graphs than those explored in other studies [15, 31, 71].

**Baseline algorithms.** We evaluate BAM's performance against other algorithms: PC [61], PC-HSIC [73], rcot and rcit [62], ccdr [3], GES [13], GIES [23], LiNGAM [60], MMPC [64], CAM [11], SAM [30], all implemented in the causal discovery toolbox [29], GLASSO [17], as implemented by [50], along with DAG-GNN [71]. Default hyperparameters as in [29] are used. For GLASSO, cross-validation for the sparsity parameter is performed as proposed by [50]. As a supervised causal discovery baseline, we use the AVICI model [42], both a pre-trained version (SCM-v0) and a version trained from scratch on the same Chebyshev data as our method, using default hyperparameters and data dimensions $d \in \{2, 5, 10, 20, 30, 40, 60, 80, 100\}$. For the number of samples $M$, which can only be chosen as a single hyperparameter value for AVICI [42] due to memory allocation constraints, we selected $M = 150$. GLASSO and MMPC are excluded from CPDAG estimation as they focus on undirected graphs. For algorithms estimating directed edges, the skeleton is computed for undirected prediction. We also compare SHD and accuracy results against a naive zero graph without edges.

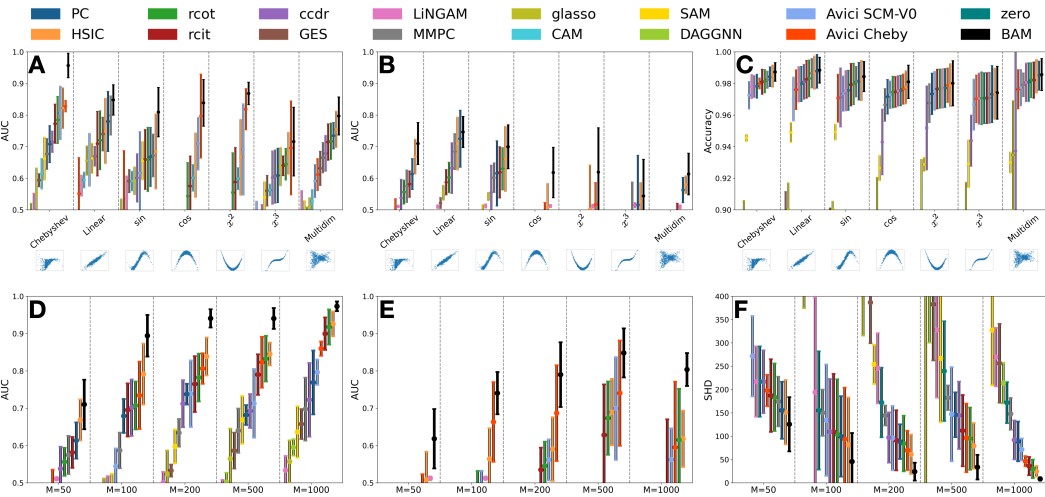

Figure 4: Undirected graph estimation results arranged from worst (left) to best (right) performance. (A, B) AUC values for different dependencies, with (A) $d = 50$, $M = 200$, and (B) $d = 100$, $M = 50$. (C) Accuracy for the same dependencies as in (B). (D, E) AUC vs. $M$ for $d = 100$ with (D) Chebyshev and (E) cosine dependencies. (F) SHD vs. $M$ for $d = 100$ with Chebyshev dependency.

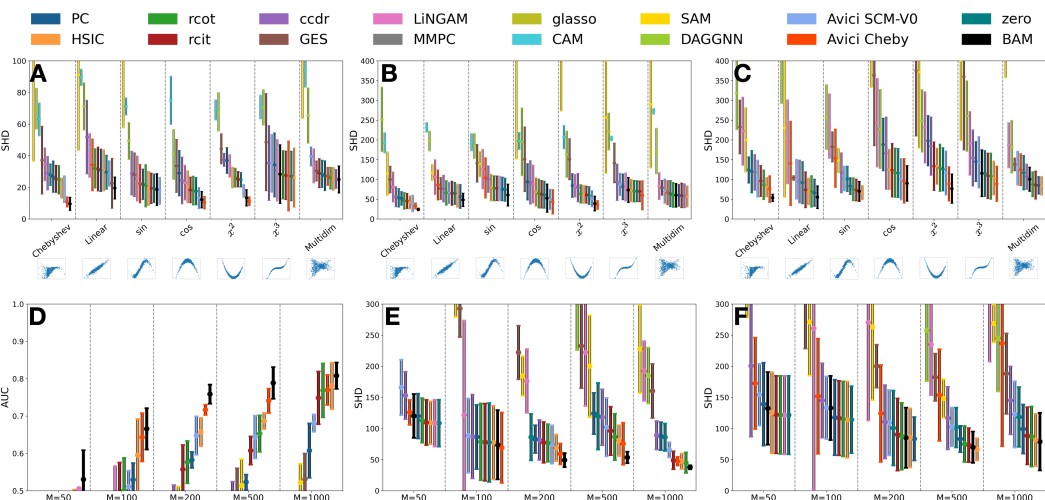

Figure 5: CPDAG estimation results arranged from worst (left) to best (right) performance. (A-C) SHD for different dependencies, with (A) $d = 20$, $M = 200$, (B) $d = 50$, $M = 200$, and (C) $d = 100$, $M = 500$. (D) AUC vs. $M$ for $d = 100$ with Chebyshev dependency. (E, F) SHD vs. $M$ for $d = 100$ with (E) Chebyshev and (F) sine dependencies.

**Performance indicators.** We assess the performance of both undirected and CPDAG graph estimations using the Area Under the Precision-Recall Curve (AUC), which is well-suited for imbalanced binary classification tasks like sparse graph detection [24]. For CPDAG estimations, we also employ the Structural Hamming Distance (SHD), a standard metric in structure learning [71, 31]. In the undirected graph estimation task, SHD is equivalent to accuracy, defined as the percentage of correctly inferred edges, which we also report. Additional Structural Intervention Distance (SID) [52] results for CPDAG estimation are provided in Appendix G.5.

**Graph estimation results.** Figure 4 showcases BAM's efficacy in undirected graph prediction. Trained on synthetic Chebyshev polynomial data, BAM consistently outperforms other methods

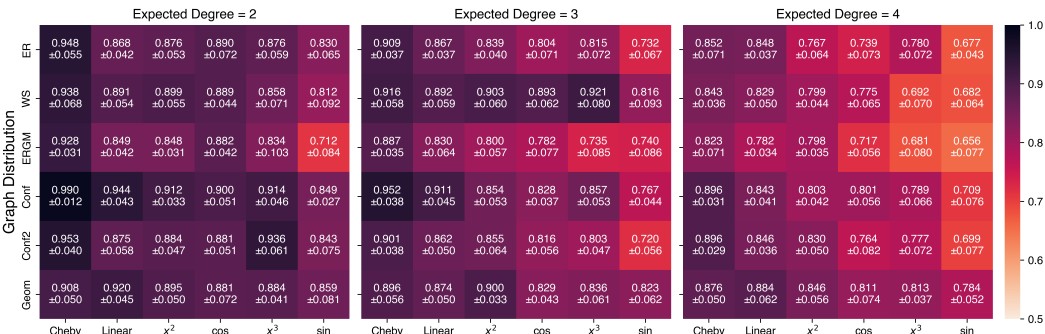

Figure 6: Heatmaps of mean AUC values with standard deviations across graph distributions and dependency functions for expected degrees 2, 3, and 4 obtained from 10 independent simulation runs. Graph distributions include Erdős-Rényi (ER), Watts-Strogatz (WS) [69], Exponential Random Graph Model (ERGM) [56], configuration models with both homogeneous (Conf) and heterogeneous (Conf2) degree distributions [47], and geometric random graphs (Geom) [51].

across various dependency relations, dimensions, and sample sizes (Figure 4 (A - C)). It excels in capturing intricate non-monotonic dependencies (Figure 4 (A, B, E) and demonstrates superior performance over Avici on the training set (Figure 4 (D, F)).

Figure 5 displays the results for CPDAG estimation tasks. In high-dimensional scenarios ($d = 100$, $M = 50$), no algorithm surpassed the baseline of a zero-graph in SHD, as shown in Appendix G.4. Therefore, our SHD analysis focuses on the low-dimensional setting shown in (A) and two moderate-dimensional ($M > d$) settings in (B) and in (C) across various dependencies. Across these scenarios, the two-step method of our algorithm remains competitive, though its advantage is less pronounced than in the task of undirected graph estimation. Panels (D) and (E) depict the AUC and SHD across varying $M$ values for the training datasets of BAM and the re-trained Avici, illustrating BAM's competitive performance in learning CPDAG-structure. Specifically, panel (E) presents SHD values for sine dependency across different $M$ values for $d = 100$, further demonstrating the robustness of BAM in non-linear settings. Additional evaluation results for more complex multivariate dependencies using random Fourier features and MLPs are provided in Appendix G.3 and Figure 9, respectively. Additional SID results for CPDAG estimation in Appendix G.5 demonstrate that our method achieves the highest performance for the best DAG of the estimated equivalence class. Compared to competing algorithms, it infers fewer edge orientations.

**Graph distribution robustness.** To demonstrate that our architecture generalizes beyond the Erdős-Rényi graphs it was trained on, we evaluate its performance on graphs generated using various random graph models. Our results, shown in Figure 6, demonstrate consistent performance across all graph distributions, suggesting that our method is robust to variations in underlying graph structure. Details about the configuration of the distributions can be found in Appendix G.6.

**Ablation study.** Table 1 presents our ablation study results, underscoring the crucial role of the bilinear layer, whose removal significantly increases loss metrics. Omitting the LogEig layer leads to a substantial deterioration in loss, indicating that direct predictions from the SPD-space are problematic. Notably, bilinear data processing alone yields relatively good results, possibly due to the embedding layer's ability to

Table 1: Ablation study results featuring loss values. Data generated under Chebyshev dependencies. "−" indicates the removal of a corresponding layer. ΔParam quantifies the difference in the number of parameters to the full model.

| model | loss ↓ | ΔParam |
|---|---|---|
| FULL | $0.173 \pm 0.007$ | |
| −BAM | $0.202 \pm 0.005$ | 80 K |
| −BAM −LogEig | $0.271 \pm 0.012$ | 100 K |
| −obs. att. | $0.189 \pm 0.006$ | 120 K |
| −obs. att. −Dense | $0.205 \pm 0.006$ | 160 K |

decode non-linearities in the data, which are lost when solely relying on covariance matrices. Further details are provided in Appendix F.2.

**Effectiveness of novel two-step approach for CPDAG estimation.** To validate our approach of identifying immoralities through the first network step, we ran additional experiments where we maintained the undirected edge estimation network but modified the second step. Specifically, we first obtained an undirected graph as the union of skeleton and moralized edges, then trained another neural network with identical hyperparameters to test each possible immorality in this graph, not just those identified by our first network. Table 2 shows that our selective testing strategy consistently out-performs this exhaustive approach across different settings, particularly for Chebyshev dependencies used in training. This suggests that conditional dependencies are more accurately identified when leveraging information from the entire graph structure, rather than examining estimated local Markov blankets independently.

Table 2: CPDAG estimation performance (AUC scores) with $M = 200$ samples. The baseline ('All Moralizations', AM) tests all possible moralizations, while BAM only tests those identified by the first network. Mean $\pm$ standard deviation over 10 runs.

| **Dependency Function** | **Model** | $d = 10$ | $d = 20$ | $d = 50$ | $d = 100$ |
|---|---|---|---|---|---|
| Chebyshev | BAM | $0.79 \pm 0.09$ | $0.85 \pm 0.05$ | $0.78 \pm 0.05$ | $0.76 \pm 0.03$ |
| | AM | $0.73 \pm 0.03$ | $0.69 \pm 0.02$ | $0.71 \pm 0.02$ | $0.70 \pm 0.01$ |
| Linear | BAM | $0.71 \pm 0.06$ | $0.72 \pm 0.02$ | $0.69 \pm 0.06$ | $0.70 \pm 0.02$ |
| | AM | $0.67 \pm 0.03$ | $0.63 \pm 0.06$ | $0.63 \pm 0.04$ | $0.68 \pm 0.03$ |
| sin | BAM | $0.69 \pm 0.16$ | $0.68 \pm 0.09$ | $0.62 \pm 0.07$ | $0.66 \pm 0.04$ |
| | AM | $0.63 \pm 0.08$ | $0.62 \pm 0.07$ | $0.60 \pm 0.06$ | $0.64 \pm 0.04$ |
| cos | BAM | $0.71 \pm 0.04$ | $0.74 \pm 0.05$ | $0.65 \pm 0.11$ | $0.59 \pm 0.04$ |
| | AM | $0.67 \pm 0.07$ | $0.64 \pm 0.05$ | $0.62 \pm 0.07$ | $0.59 \pm 0.05$ |
| $x^2$ | BAM | $0.77 \pm 0.05$ | $0.77 \pm 0.09$ | $0.71 \pm 0.08$ | $0.63 \pm 0.11$ |
| | AM | $0.68 \pm 0.06$ | $0.64 \pm 0.05$ | $0.64 \pm 0.05$ | $0.61 \pm 0.05$ |
| $x^3$ | BAM | $0.56 \pm 0.03$ | $0.55 \pm 0.17$ | $0.54 \pm 0.09$ | $0.59 \pm 0.06$ |
| | AM | $0.58 \pm 0.05$ | $0.50 \pm 0.10$ | $0.56 \pm 0.09$ | $0.57 \pm 0.05$ |

**Efficiency.** Training our neural network takes approximately 6 hours on an A-100 GPU with 81,920 MiB of graphical memory, while inference typically requires less than a few seconds and can be run on a normal computer. The overall memory complexity of our proposed method is $O(CMd + Cd^2 + Md^2 + M^2d + C^2)$. The runtime complexity of our approach is $O(C^2Md + CMd^2 + CM^2d + C^2d^2 + Cd^3)$. We present a detailed breakdown of the single components in Appendix H. Empirical evaluations show that our model achieves significantly lower computational times compared to most unsupervised approaches, as shown in Appendix G.7.

## 5 Conclusion

In this study, we introduced a novel neural network model for supervised graph structure learning that addresses identifiability issues in observational data through a two-step approach. Our approach extends the classical GLASSO paradigm to the neural network era by incorporating a novel bilinear attention mechanism operating in both Euclidean and SPD spaces, effectively learning shape-invariant transformations directly from data rather than relying on pre-derived mathematical operations. The trained model effectively detects causal relationships and generalizes well across different functional forms of nonlinear dependencies, while requiring only a single forward pass for inference. Comprehensive empirical evaluations demonstrate that our approach consistently outperforms state-of-the-art methods across diverse scenarios. Future directions include extending our approach to handle more complex settings and investigating its performance under various noise and confounding conditions, as well as scaling through local attention mechanisms. Limitations and future directions are further discussed in Appendix J.

## Acknowledgments and funding disclosure

This project (HA project no.: 1304/22-09) is funded by the State of Hesse and the HOLM funding programme as part of the 'Innovations in the field of logistics and mobility' initiative of the Hessian Ministry of Economics, Energy, Transport and Housing. Additional funding was provided by the Federal Ministry of Education and Research (BMBF, project KI4ELM - 01IS24022C) and the European Union's Horizon 2020 programme iPC (individualizedPaediatricCure) project under grant agreement No. 826121.

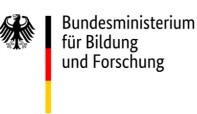 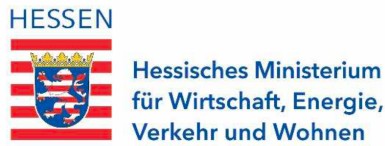 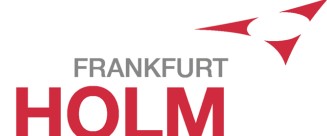

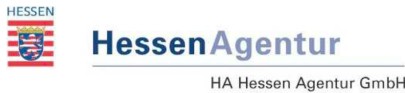 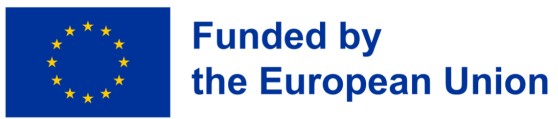

The computational work was supported by the High-Performance Computing facility Lichtenberg at the NHR Centers NHR4CES at TU Darmstadt.

**Competing interests:** The authors declare no competing interests.

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

# A  Properties and theorems regarding the custom softmax function

Similar to the standard softmax, our softmax for an SPD matrix returns positive values summing to $d$, but unlike the standard softmax, the rows do not sum to 1. We demonstrate that our modified softmax function $\widetilde{\sigma}$ regularizes the eigenvalues:

**Theorem 1.** *For any $S \in \mathcal{S}_{\succeq}^{d \times d}$, the largest eigenvalue of $\widetilde{\sigma}(S)$ is 1.*

*Proof.* Let $\widetilde{S} := \exp[S]$. By similarity transformation, the eigenvalues of $\widetilde{\sigma}(S)$ are equal to the eigenvalues of $\widetilde{S}\Lambda(S)$. It holds that $\widetilde{S}\Lambda(S)\widetilde{S}\mathbf{1} = \widetilde{S}\mathbf{1}$, which demonstrates that $\widetilde{S}\mathbf{1}$ is the Perron eigenvector corresponding to the eigenvalue 1. The assertion now follows from the Perron-Frobenius theorem. $\qquad\square$

**Proposition 2.** *The custom softmax $\widetilde{\sigma}$ is invariant to additive shifting, i.e., $\widetilde{\sigma}(S + \alpha) = \widetilde{\sigma}(S)$ for each $\alpha \in \mathbb{R}$, $S \in \mathcal{S}_{\succeq}^{d \times d}$.*

Proposition 2 shows that $\widetilde{\sigma}$, unlike standard softmax, does not need a scaling constant, and it can easily manage exploding $\exp$ values via maximum-value scaling.

# B  Architectural details

## B.1  Observational attention

This layer accepts an input $X \in \mathbb{R}^{M \times d \times C}$, from which it generates keys $K = XW_K \in \mathbb{R}^{M \times d \times c}$, queries $Q = XW_Q \in \mathbb{R}^{M \times d \times c}$, and values $V = XW_V \in \mathbb{R}^{M \times d \times C}$, with $W_K, W_Q \in \mathbb{R}^{C \times c}$, and $W_V \in \mathbb{R}^{C \times C}$.

For each $m = 1, \ldots, M$, keys and queries are combined in parallel along the inner axis, leading to

$$K \odot Q := \big(K_{m,\cdot,\cdot} Q_{m,\cdot,\cdot}\big)_{m=1,\ldots,M} \in \mathbb{R}^{M \times d \times d}. \tag{4}$$

This results in the attention weights

$$A = \sigma\left(\frac{K \odot Q^{T(1,3,2)}}{\sqrt{c}}\right) \in \mathbb{R}^{M \times d \times d},$$

where $T(\mathrm{perm})$ denotes a permutation of the axes according to a permutation $\mathrm{perm}$ and $\sigma$ is the softmax operation along the last axis. For example, if $\mathrm{perm} = [1, 3, 2]$, the operation would rearrange the second and third axes of the tensor while the first axis stays unchanged. Finally, for each $m$, the output is computed as

$$H = A \odot V \in \mathbb{R}^{M \times d \times C}. \tag{5}$$

## B.2  SPD activation function

To obtain an activation function for the SPD net, we leverage the following theorem, a direct consequence of (Theorem 4.11 Guillot and Rajaratnam, 2015), which is based on the work of [57]:

**Theorem 3.** *Any continuous function from $C([-1, 1], [-1, 1])$ acting elementwise on a matrix preserves positive definiteness if it can be expressed as a series $f : [-1, 1] \to [-1, 1]$ with*

$$f(x) = \sum_{k=1}^{\infty} x^k w_k,$$

$$\text{subject to} \quad \sum_{k=1}^{\infty} w_k \leq 1 \quad \text{with} \quad w_k \geq 0, \quad \forall k \in \mathbb{N}.$$

We employ this theorem to construct an activation function for the SPD neural network, using a relatively small maximal polynomial degree value of $N_{\max} = 3$. This function employs trainable weights $w_k$ and is applied after 'correlation normalization', i.e., the conversion of covariance matrices into correlation matrices. Trainable activation functions using low-degree Taylor polynomials were also proposed in [14] for general neural networks, not focusing on SPD data. Additionally, [2] provides various types of trainable activation functions for Euclidean neural networks.

### B.3 Residual connections and normalization.

Normalization is critical in attention networks, with various methodologies available [25, 5]. We adopt normalization and a residual connection [25] on the input of all attention layers, expressed as $\boldsymbol{S} + \text{Attention}(\text{LayerNorm}(\boldsymbol{S}))$. Particularly when $M < d$, the residual connection can enhance the rank of covariance matrices, so full-rank representations can be attained even for under-determined problems. For learnable residual scaling, we utilize recently established methods [6].

Within the SPD manifold, we employ correlation normalization alongside residual connections. This approach preserves positive definiteness and corresponds intuitively to standard normalization in Euclidean space.

### B.4 Multiple heads

In all attention layers, we employ multihead attention, a process that divides the input tensor along the channel axis into several smaller tensors. Each of these is then subjected to attention independently.

### B.5 Weight initialization

For initialization of the $C \times C$ weight matrices $\boldsymbol{W}^+$ in each bilinear attention layer, we draw initial weights from $U\left(0, \frac{2}{C}\right)$. Given a tensor of SPD matrices $\boldsymbol{\Sigma} \in \mathcal{S}_{\succeq}^{d \times d \times C}$, in expectation, the diagonal entries in each of the $d \times d$ matrices in $\boldsymbol{\Sigma}\boldsymbol{W}^+$ should stay within a comparable range, while off-diagonal entries are expected to be drawn to zero due to the symmetric distribution of positive and negative values, resulting in perturbed identity matrices for keys and queries.

## C Theoretical foundation for the three-class edge classification problem

Building upon the foundational principles of Markov and faithfulness [33], we demonstrate that the three-class classification problem can be theoretically deduced from the distribution of the nodes by examining the following independence relations: If there is an directed edge from node $X$ to $Y$ in the DAG, then $X$ and $Y$ are dependent given any set from the power set of the other nodes, i.e.:

$$X \to Y \implies X \not\perp\!\!\!\perp Y \mid \boldsymbol{C} \quad \forall \boldsymbol{C} \in \mathcal{P}(\mathcal{V} \setminus \{X, Y\}).$$

For an immorality between $X$ and $Y$, there exists a set of nodes $\boldsymbol{C} \in \mathcal{P}(\mathcal{V} \setminus \{X, Y\})$ within the power set of all other nodes such that $X$ and $Y$ are conditionally independent given this set (e.g., the set of all common ancestors of $X$ and $Y$, or the set of all parents of $X$ or $Y$), but dependent given all other nodes in the graph, i.e.:

$$X \to Z \leftarrow Y, X \not\leftarrow Y, X \not\to Y \implies \exists \boldsymbol{C} \in \mathcal{P}(\mathcal{V} \setminus \{X, Y\}) : X \perp\!\!\!\perp Y \mid \boldsymbol{C}, X \not\perp\!\!\!\perp Y \mid \mathcal{V} \setminus \{X, Y\}.$$

If there is no edge between $X$ and $Y$, and if $X$ and $Y$ do not have a common child, then $X$ and $Y$ are conditionally independent given all other nodes, i.e.,

$$X \not\leftarrow Y, X \not\to Y, \quad \nexists Z : X \to Z \leftarrow Y \implies X \perp\!\!\!\perp Y \mid \mathcal{V} \setminus \{X, Y\}.$$

Note that testing for the no-edge class is cost-effective, as one only needs to test for a single set, rather than checking for any $Z \in \mathcal{V} \setminus \{X, Y\}$ if there is a v-structure $X \to Z \leftarrow Y$. Testing for a sepset to differentiate between the skeleton and moralized edge classes is more intricate. The neural network is tasked with learning an approximation for this distinction.

## D Details of the CPDAG estimation model

To estimate a CPDAG from the graph skeleton, along with the set of immoralities between pairs of nodes, we test each estimated immorality to determine which potential common child nodes are indeed common children. The parent nodes, denoted by pa, are the nodes between which an immorality was first estimated. Potential common child nodes, denoted by cc, are nodes that have an edge to both parent nodes. Neighbor nodes, denoted by ne, are nodes that have one edge to exactly one parent.

We take the columns $\boldsymbol{x}_{\mathrm{pa}} \in \mathbb{R}^{M \times 2}, \boldsymbol{x}_{\mathrm{cc}} \in \mathbb{R}^{M \times |\mathrm{cc}|}, \boldsymbol{x}_{\mathrm{ne}} \in \mathbb{R}^{M \times |\mathrm{ne}|}$ of the data matrix $\boldsymbol{X}$ corresponding to pa, cc, ne as inputs for the neural network. Each of these submatrices undergoes a dimensionality expansion to $\widetilde{\boldsymbol{x}}_{\mathrm{pa}} \in \mathbb{R}^{M \times 2 \times 1}, \widetilde{\boldsymbol{x}}_{\mathrm{cc}} \in \mathbb{R}^{M \times |\mathrm{cc}| \times 1}, \widetilde{\boldsymbol{x}}_{\mathrm{ne}} \in \mathbb{R}^{M \times |\mathrm{ne}| \times 1}$. Next, three separate feed-forward subnetworks $l_{\mathrm{pa}}, l_{\mathrm{cc}}, l_{\mathrm{ne}}$ are applied to the three inputs $\boldsymbol{x}_{\mathrm{pa}}, \boldsymbol{x}_{\mathrm{cc}}, \boldsymbol{x}_{\mathrm{ne}}$. Each of these layers has the same architecture: For an input $\widetilde{\boldsymbol{x}} \in \mathbb{R}^{M \times d \times 1}$, weight matrices $\boldsymbol{W}_1 \in \mathbb{R}^{1 \times C}$ and $\widetilde{\boldsymbol{W}}_1 \in \mathbb{R}^{C \times C}$, together with a bias vector $\boldsymbol{b}_1 \in \mathbb{R}^C$ are used to embed $\widetilde{\boldsymbol{x}}$ to

$$\boldsymbol{h}_1 = \tanh(\widetilde{\boldsymbol{x}} \boldsymbol{W}_1 + \boldsymbol{b}_1) \widetilde{\boldsymbol{W}}_1 \in \mathbb{R}^{M \times d \times C}$$

Then, two residual layers of the form

$$\boldsymbol{h}_{i+1} = \boldsymbol{h}_i + \tanh(\boldsymbol{h}_i \boldsymbol{W}_i + \boldsymbol{b}_i) \widetilde{\boldsymbol{W}}_i$$

with $\boldsymbol{W}_i \in \mathbb{R}^{C \times C}, \widetilde{\boldsymbol{W}}_i \in R^{C \times C}, \boldsymbol{b}_i \in \mathbb{R}^C, i = 1, 2$ are applied to obtain representations $\boldsymbol{h}_{\mathrm{pa}} \in \mathbb{R}^{M \times 2 \times C}, \boldsymbol{h}_{\mathrm{cc}} \in \mathbb{R}^{M \times |\mathrm{cc}| \times C}, \boldsymbol{h}_{\mathrm{ne}} \in \mathbb{R}^{M \times |\mathrm{ne}| \times C}$. For the addition of the bias terms, broadcasting is used, i.e.,

$$\boldsymbol{b} \in \mathbb{R}^C \mapsto \widetilde{\boldsymbol{b}} \in \mathbb{R}^{M \times d \times C} \text{ with } \widetilde{\boldsymbol{b}}_{m,l,c} = \boldsymbol{b}_c \quad m = 1, \ldots, M, \ l = 1 \ldots, d, \ c = 1, \ldots, C.$$

Now, the representations are concatenated along the dimension axis to obtain a $M \times (2 + |\mathrm{cc}| + |\mathrm{ne}|) \times C$ tensor. We use $d = 2 + |\mathrm{cc}| + |\mathrm{ne}|$. We employ the same observation-to-dependency network as before, but instead of using softmax on the output of the LogEig-Layer, we use mean-pooling

$$\boldsymbol{X} \mapsto \left( \frac{1}{d} \boldsymbol{X}^{T(3,2,1)} \mathbf{1}_d \right)^T$$

to inflate one of the variable axes of the $\mathbb{R}^{d \times d \times C}$ output of LogEig to obtain a $\mathbb{R}^{d \times C}$ batch of $C$ vectors of dimension $d$. After applying a dense $C \times 1$ layer, we obtain a vector of length $d$. Now, the entries corresponding to the potential common children can be sliced out and backpropagated for training.

The network architecture is shown in figure 7.

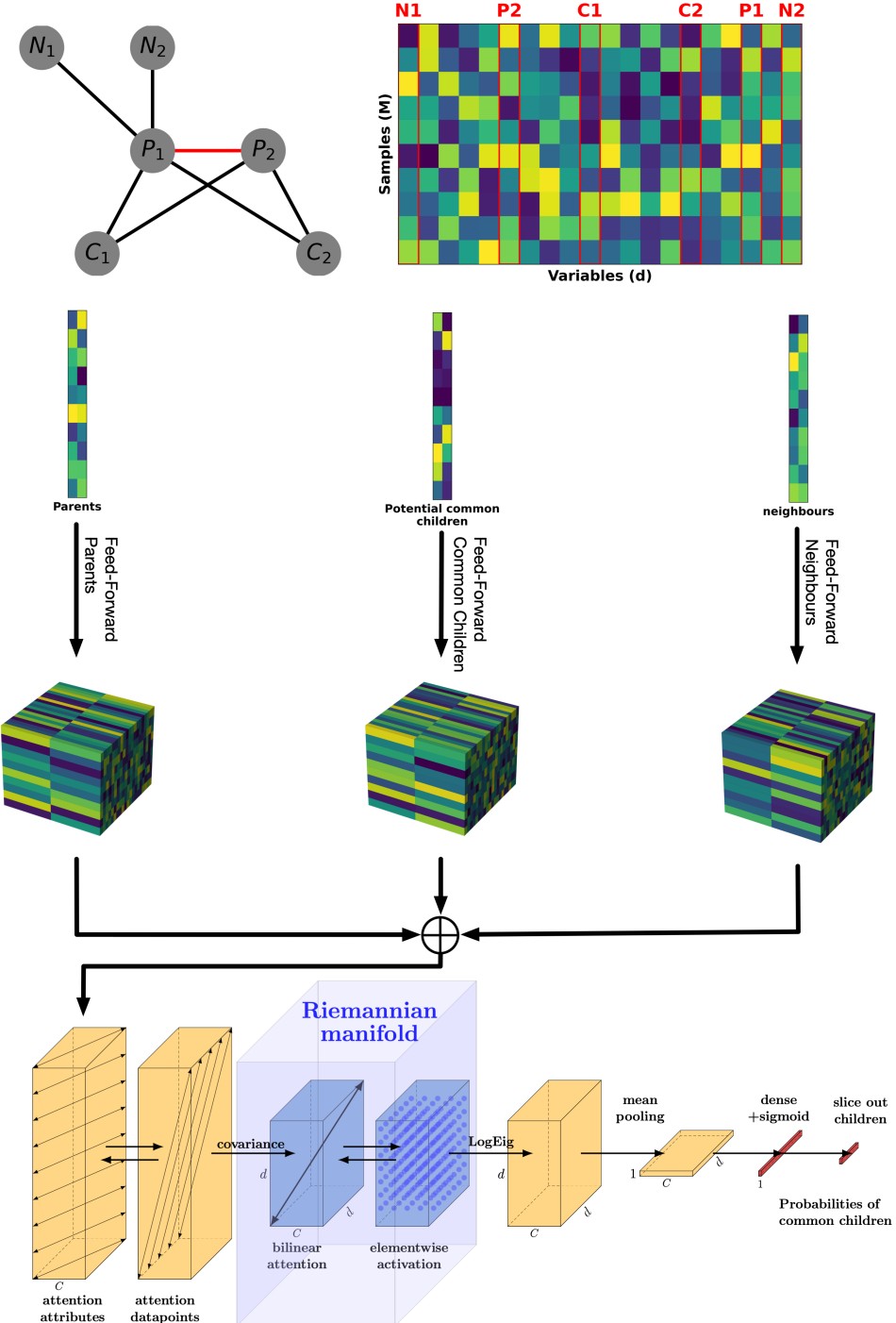

Figure 7: CPDAG Estimation Architecture: In the graph, black edges represent undirected connections, whereas the red edge signifies an immorality. Columns in the data matrix corresponding to parent nodes, potential common children, and neighboring nodes are taken as input for the neural network and are processed through three distinct feed-forward networks. The embeddings are concatenated along the variable axis and processed through a network with data matrix attention, bilinear attention, and a LogEig layer. Mean pooling and sigmoid activation are applied to output probabilities for potential common children.

# E Parameterization of the SEM

## E.1 Chebyshev polynomials for training

For the Chebyshev polynomial, we utilize the following parameterization:

$$
f_v(\boldsymbol{x}_{\mathrm{pa}}, \epsilon_1, \epsilon_2) = \sum_{w \in \mathrm{pa}_{\mathcal{G}}(v)} \beta_w \sum_{n=1}^{r} \alpha_n T_n(x_w) + \epsilon_1
$$

$$
+ \alpha_{\mathrm{m}} \left( \sum_{s,t \in \mathrm{pa}(\mathcal{G}), s<t} \delta_{s,t} T_{\mathrm{m}}(x_s, x_t) + \sum_{w \in \mathrm{pa}(\mathcal{G})} T_{\mathrm{m}}(x_w, \epsilon_2) \right) \quad \forall v \in \mathcal{V},
$$

where $r = 5$ is the degree, and $T_n$ denotes the Chebyshev polynomials of the first kind (scaled for the input to have a maximum absolute value of 1), and bivariate polynomials are given by:

$$
T_{\mathrm{m}}(x, y) := \frac{(x - \mu_x)(y - \mu_y)}{(1 + |\mu_x|)(1 + |\mu_y|)} \tag{6}
$$

where m in the index stands for "multidimensional".

Here, $\mu_x \sim U[-1, 1]$, $\mu_y \sim U[-1, 1]$, and the coefficients $\alpha_1, \dots, \alpha_r, \alpha_{\mathrm{m}}$ are calculated from $\alpha_i = \frac{\widetilde{\alpha}_i}{\sum_n \widetilde{\alpha}_n + \widetilde{\alpha}_{\mathrm{m}}}$ for $i = 1, \dots, r$, with $\widetilde{\alpha}_i = \frac{\gamma_i}{i!}$, $\gamma_i \sim U[-1, 1]$, and $\widetilde{\alpha}_{\mathrm{m}} \sim U[-1, 1]$. $\beta_w = \frac{\widetilde{\beta}_w}{|\mathrm{pa}|}$ with $\widetilde{\beta}_w \sim U[0.7, 1.3]$. $\delta_{s,t}$ are random weights with $\delta_{s,t} = \frac{\widetilde{\delta}_{s,t}}{\sum_{s,t \in \mathrm{pa}(\mathcal{G}), s<t} |\widetilde{\delta}_{s,t}|}$, $\widetilde{\delta}_{s,t} \sim U[-1, 1]$

This parameterization is motivated by the observation that for a smooth function—where higher-order derivatives are not significantly larger than the lower-order ones—the coefficients of the Chebyshev approximation decrease in a factorial manner. This provides a rationale for training the neural network on 'typical' smooth functional dependencies. Furthermore, this suggests that using Chebyshev polynomials of degree $r = 5$ is not a significant limitation, as the coefficients of higher orders are already negligibly small.

In order to prevent the values from exploding and to properly account for the common domain of Chebyshev polynomials, we implement several measures. Firstly, we scale each input to the SEM by the maximum value within the batch. Secondly, we standardize all variables; we subtract the mean and divide by the standard deviation for each batch. This ensures the variables are both centered and scaled. To further improve stability and robustness of our model, we introduce thresholds for any absolute values exceeding 5, thereby mitigating the potential impact of outliers.

## E.2 Gaussian mixture error terms

The additive error term $\epsilon_1$ follows a Gaussian mixture distribution. We randomly determine the number of components $L \sim U\{1, \dots, 5\}$ from a discrete uniform distribution. Each component has randomly assigned parameters for the means $\mu_l \sim U[-1, 1]$, standard deviations $\sigma_l \sim U[0.05, 1]$, and weights $\widetilde{w}_l \sim U[0.3, 1]$, $w_l = \frac{\widetilde{w}_l}{\sum_l \widetilde{w}_l}$ such that $\epsilon_1 \sim \sum_{l=1}^{L} w_l N(\mu_l, \sigma_l^2)$. The multiplicative error term $\epsilon_2$ is uniformly distributed, with $\epsilon_2 \sim U[-1, 1]$. We scale the error to zero mean and variance 1 after simulation.

## E.3 Testing dependencies

To create testing data, we create synthetic data according to an SEM equipped with different dependency function, while the error term follows a Gaussian mixture distribution as before. We use the following dependencies for testing: Chebyshev, linear, sine, cosine, $x^2$, $x^3$, multidimensional multiplicative dependency. The Chebyshev-dependency used was the same as in the training procedure. We used the following testing dependency functions: $x$, $\sin(x)$, $\cos(x)$, $x^2$, $x^3$ as $g(x)$ in

$$
f_{\mathrm{test}}(\boldsymbol{x}_{\mathrm{pa}}, \varepsilon) = \sum_{w \in \mathrm{pa}(\mathcal{G})} \alpha_w g(x_w) + \epsilon.
$$

For the multi-dimensional multiplicative test dependency, we used

$$f_v(\boldsymbol{x}_{\mathrm{pa}}, \epsilon_1, \epsilon_2) = \alpha_{\mathrm{m}} \left( \sum_{s,t \in \mathrm{pa}(\mathcal{G}), s<t} \delta_{s,t} T_{\mathrm{m}}(x_s, x_t) + \sum_{w \in \mathrm{pa}(\mathcal{G})} T_{\mathrm{m}}(x_w, \epsilon_2) \right) \quad \forall v \in \mathcal{V},$$

with

$$T_{\mathrm{m}}(x, y) := \frac{(x - \mu_x)(y - \mu_y)}{(1 + |\mu_x|)(1 + |\mu_y|)}$$

with $\mu_x \sim U[-1, 1]$, $\mu_y \sim U[-1, 1]$, $\delta_{s,t} = \frac{\widetilde{\delta}_{s,t}}{\sum_{s,t \in \mathrm{pa}(\mathcal{G}), s<t} |\widetilde{\delta}_{s,t}|}$, $\widetilde{\delta}_{s,t} \sim U[-1, 1]$.

# F  Training details

## F.1  Model hyperparameters

The implementation was performed using TensorFlow [1]. We employed the ADAM optimizer by Kingma and Ba [32]. For training, we used the hyperparameters stated in Table 3:

Table 3: Hyperparameters for the undirected graph estimation

| Hyperparameter | Value |
|---|---|
| **Layer Parameters** | |
| Number of channels $C$ | 100 |
| Number of inner channels $c$ | 100 |
| Maximal degree activation function | 3 |
| Attention heads | 5 |
| **Number of layers** | |
| Attention between attributes | 10 |
| Attention between samples | 10 |
| $C \times C$ dense observational layers | 10 |
| bilinear attention + SPD activation | 10 |
| **Training Schedule** | |
| epochs | 1000 |
| samples per epoch | 128 |
| Initial learning rate | 0.0005 |
| Learning rate decrease factor | $\left(\frac{1}{10}\right)^{1/500}$ |
| Minibatchsize | 1 |

Table 4: Hyperparameters of the CPDAG estimation model

| Hyperparameter | Value |
|---|---|
| **Layer Parameters** | |
| Number of channels $C$ | 100 |
| Number of inner channels $c$ | 100 |
| Maximal degree activation function | 3 |
| Attention heads | 5 |
| **Number of layers** | |
| Attention between attributes | 10 |
| Attention between samples | 10 |
| $C \times C$ dense observational layers | 10 |
| bilinear attention + SPD activation | 10 |
| **Training Schedule** | |
| epochs | 1000 |
| matrices per epoch | 1 |
| Initial learning rate | 0.0005 |
| Learning rate decrease factor | $\left(\frac{1}{10}\right)^{1/1000}$ |
| Minibatchsize | 1 |

Additionally, we generated data with a random number of samples $M \sim U\{50, 51, \ldots, 1000\}$ and a random variable dimension $d \sim U\{10, 11, \ldots, 100\}$.

## F.2  Ablation study setting

In the ablation study, we evaluate the performance of the full model in comparison to models with reduced complexities. The full model is comprised of two attention between attributes layers, two attention between samples layers, two dense layers, and four bilinear layers equipped with SPD activation functions. This setup maintains parity between the number of attention layers operating on observational data and those focusing on covariance data, while also ensuring a comparable parameter count across different configurations. All models in the study utilize $C = c = 100$ channels and are trained over 500 epochs, with each epoch comprising 128 data matrix / adjacency label pairs. Again, we generated data with a random number of samples $M \sim U\{50, 51, \ldots, 1000\}$ and a random variable dimension $d \sim U\{10, 11, \ldots, 100\}$.

## F.3  Loss function for the three-class edge classification problem

We employ the categorical cross-entropy loss function for classifying edges into one of three categories: *no-edge*, *skeleton edge*, and *moralized edge*. Additionally, to enforce the condition that a moral edge between nodes $X$ and $Y$ should only be predicted if there is a potential common child $Z$ (i.e., $X - Z$ and $Z - Y$), we introduce a penalty term, $\mathcal{L}_{\mathrm{p}}$.

The overall loss function is defined as $\mathcal{L}_{\mathrm{b}} + \mathcal{L}_{\mathrm{c}} + \mathcal{L}_{\mathrm{p}}$. Here, $\mathcal{L}_{\mathrm{c}}$ is the categorical crossentropy of the three categories given by

$$\mathcal{L}_{\mathrm{c}}(\boldsymbol{A}, \widehat{\boldsymbol{A}}) := H(\boldsymbol{A}, \widehat{\boldsymbol{A}}) = -\sum_{i=2}^{d}\sum_{j=1}^{i-1}\sum_{c=1}^{3} A_{i,j,c} \log(\widehat{A}_{i,j,c})$$

denotes the categorical crossentropy of the three categories *no-edge*, *skeleton edge*, and *moralized edge* with

$$\boldsymbol{A} \in \mathbb{R}^{d \times d \times 3} \quad \text{with} \quad A_{i,j,c} = \begin{cases} 1 & \text{if } (i,j) \text{ is in category } c \text{ in the ground-truth DAG} \\ 0 & \text{else} \end{cases}$$

and $\widehat{A}_{i,j,c}$ denotes the estimation by the algorithm on it. $\boldsymbol{A}$ and $\widehat{\boldsymbol{A}}$ are symmetric along its first two axes, i.e., $A_{i,j,c} = A_{j,i,c}$, $\quad i,j = 1, \ldots, d$, $\quad c = 1, \ldots, 3$.

$\mathcal{L}_{\mathrm{b}}$ denotes the binary loss of no-edge vs. any edge present (present edges = skeleton edges $\cup$ moralized edges):

$$\mathcal{L}_{\mathrm{b}}(\boldsymbol{A}^{(b)}, \widehat{\boldsymbol{A}}^{(b)}) := H(\boldsymbol{A}^{(b)}, \widehat{\boldsymbol{A}}^{(b)}) = -\sum_{i=2}^{d} \sum_{j=1}^{i} \left[ A_{i,j}^{(b)} \log(\widehat{A}_{i,j}^{(b)}) + (1 - A_{i,j}^{(b)}) \log(1 - \widehat{A}_{i,j}^{(b)}) \right]$$

with

$$\boldsymbol{A}^{(b)} \in \mathbb{R}^{d \times d} \quad \text{with} \quad A_{i,j}^{(b)} = \begin{cases} 1 & \text{if an edge is estimated between } i \text{ and } j \\ 0 & \text{else} \end{cases}$$

being the adjacency matrix of no-edge vs. (direct edge $\cup$ moralized edge).

The penalty term, $\mathcal{L}_{\mathrm{p}}$ is defined as:

$$\mathcal{L}_{\mathrm{p}}(\widehat{\boldsymbol{A}}) := \max\left( \widehat{\boldsymbol{A}}_3 - [\widehat{\boldsymbol{A}}_2 \widehat{\boldsymbol{A}}_2]^{0.5}, 0 \right).$$

$\widehat{\boldsymbol{A}}_1, \widehat{\boldsymbol{A}}_2, \widehat{\boldsymbol{A}}_3 \in \mathbb{R}^{d \times d}$ are estimates of $\boldsymbol{A}_{\cdot,\cdot,1}$, $\boldsymbol{A}_{\cdot,\cdot,2}$, and $\boldsymbol{A}_{\cdot,\cdot,3}$ by the algorithm respectively. This term penalizes the prediction of a moralized edge in the absence of potential common child edges. The square root operation is applied element-wise.

# G   Further experiments

## G.1   Error bars

In our experiments, each algorithm was evaluated on five distinct trials, each involving a unique data matrix and corresponding ground-truth graph. The bars in the figures represent the mean performance values across these trials, while the error bars indicate the standard deviation. It is important to note that the observed variability, manifested as relatively large error bars, is predominantly due to the random sampling of graph degrees, which has a substantial influence on estimation accuracy. Despite this inherent variability, the comparison across algorithms remains valid, as each algorithm is tested for the same graphs. Therefore, the magnitude of the error bars should not be interpreted as undermining the reliability of our findings.

## G.2   Additional results on undirected graph estimation

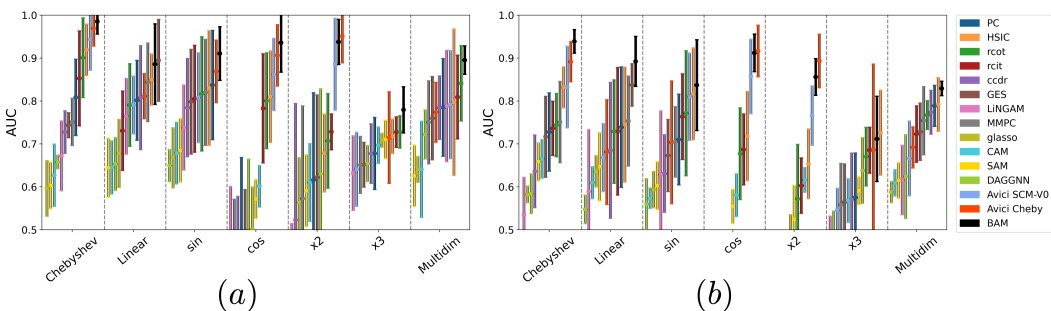

Figure 8: AUC values for undirected graph estimation in low-dimensional regimes: (a) $d = 10$, $M = 200$ (b) $d = 20$, $M = 500$.

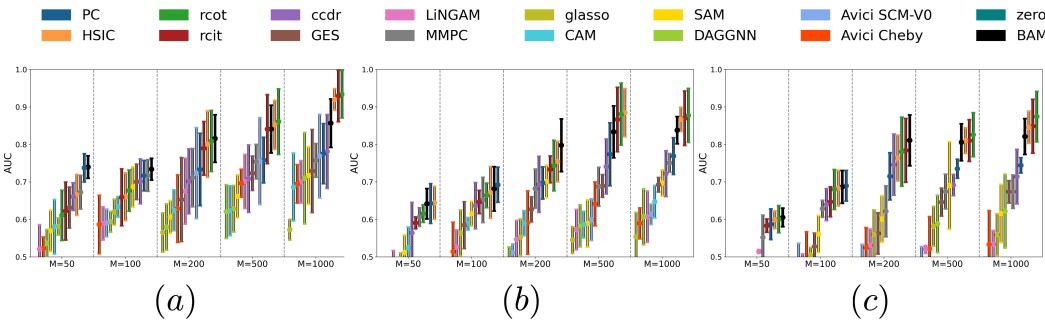

Figure 9: AUC values for undirected graph estimation for random MLP dependency: (a): $d = 20$, (b): $d = 50$, (c): $d = 100$

Figures 8 and 9 provide supplemental data on the task of undirected graph estimation. Figures 8 showcases performance in low-dimensional settings characterized by $d = 10$, $M = 200$ and $d = 20$, $M = 500$. In this scenario, our method (BAM) also outperforms competing graph inference algorithms. To further assess its capability to recognize multidimensional dependencies, we extended our tests to cases where the dependency function within the SEM is modeled via a randomly initialized multilayer perceptron (MLP) with a random number of layers $\sim \mathcal{U}\{1, \ldots, 5\}$, a random number of hidden layers $\sim \mathcal{U}\{4, 64\}$, and relu or tanh activation with probability $0.5$ each. Despite these complexities, our algorithm maintained state-of-the-art performance, delivering AUC scores competitive to the top-performing existing methods such as depicted in Figure 9.

## G.3 Additional results on random Fourier feature dependencies

We evaluate our method's performance on more complex dependency structures using random Fourier features, which represent multidimensional non-linear relationships between variables. For undirected graph estimation (Figure 10 A-C), our approach demonstrates strong performance across different graph sizes and evaluation metrics, particularly outperforming baselines for most sample sizes. The performance slightly decreases for very large sample sizes ($M = 500, 1000$), potentially due to overfitting to the training distribution when encountering dependencies that deviate significantly from the training patterns. For CPDAG estimation (Figure 10 D-F), while our method shows reasonable performance as measured by AUC and SID, there remains room for improvement in handling multidimensional dependencies. These results complement our findings on MLP dependencies (Figure 9), indicating that our approach can effectively handle various types of non-linear relationships while highlighting specific scenarios for future enhancements.

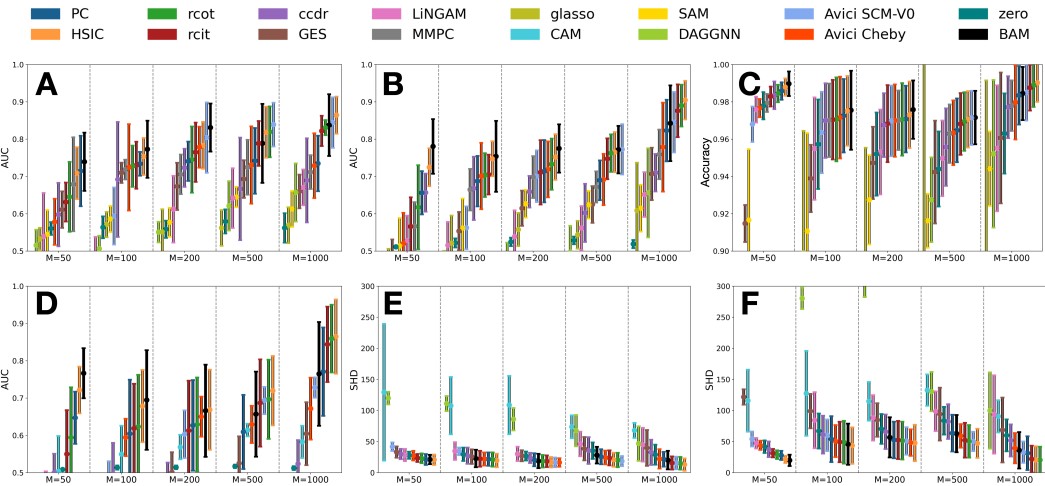

Figure 10: Additional computational results using random Fourier feature dependency. (A, B) AUC for undirected graph estimation with (A) $d = 20$, $M = 200$ and (B) $d = 50$, $M = 200$. (C) Accuracy for undirected graph estimation with $d = 50$, $M = 200$. (D) AUC for CPDAG estimation with $d = 50$, $M = 200$. (E, F) SID for CPDAG estimation with (E) $d = 20$, $M = 200$ and (F) $d = 50$, $M = 200$.

## G.4 Additional results on CPDAG estimation

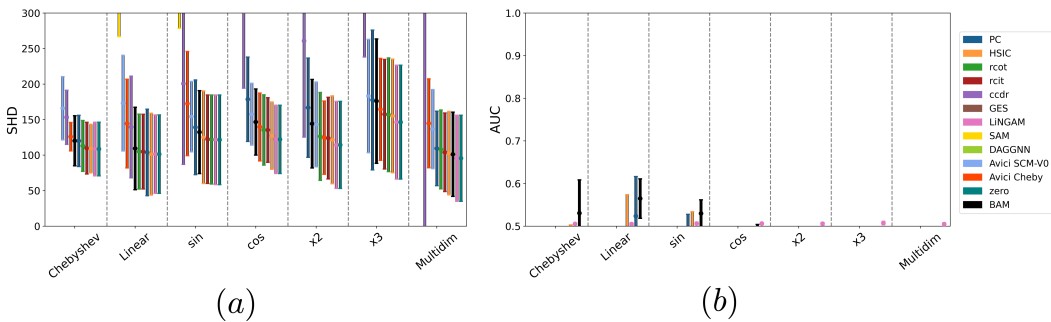

Figure 11: (a) SHD values for the high-dimensional CPDAG estimation $d = 100$, $M = 50$. (b) AUC values for the high-dimensional CPDAG estimation $d = 100$, $M = 50$.

CPDAG estimation in high-dimensional settings presents significant challenges. In the specific case of $d = 100$ and $M = 50$, none of the algorithms we evaluated could outperform a zero-graph (i.e., a

graph with no edges) baseline in terms of Structural Hamming Distance (SHD). The results, depicted in Figure 11 (a), substantiate this observation and suggest that CPDAG estimation remains a difficult problem under these conditions, at least with our chosen graph density setup. Given the complexities encountered in high-dimensional contexts, our analysis primarily emphasizes the evaluation of AUC, as illustrated in Figure 11(b). In this evaluation, only the BAM and PC algorithms demonstrated AUC values exceeding $0.5$ in certain instances. However, the AUC metrics remain low. This underscores the utility of undirected graph methods for high-dimensional ($d > M$) problems for such problems, as directed approaches may not only be inefficient but also risk yielding misleading interpretations.

### G.5 Additional results regarding SID metrics

We performed additional evaluations using the structural intervention distance (SID) metric for CPDAG estimation and graph dimensions of $d = 10$, as shown in Figure 12. For higher graph dimensions, the implementation of the SID metric within the causal discovery toolbox was leading to computational errors due to instability and high complexity of the evaluation metric. The minimal and maximal SID values, representing the best and worst DAGs within the equivalence classes, are shown. Our method has the best performance for the best DAG of the estimated equivalence class. However, the results were less favorable for the worst DAG configurations. These findings suggest that our model estimates a broad equivalence class and tends to be conservative in the directionality of edges.

### G.6 Results with varying graph distributions

We investigate the robustness of our method to different graph distributions by testing various distributions in combination with different dependency functions. Specifically, we consider the Watts-Strogatz (WS) graph [69], the Exponential Random Graph Model (ERGM) [56], the configuration model [47], and the geometric random graph [51] for generating graph structures. We report the area under the precision recall curve (AUC) values for the undirected graph estimation task for the case $M = 200, d = 20$.

The WS graph produces small-world graphs with high clustering and short average path lengths by randomly rewiring edges of a regular ring lattice. We use the implementation in the NetworkX package [22] to create WS graphs with a specified expected degree. The rewiring probability $p$ is sampled uniformly from 0.1 to 0.9 to cover a wide range of graph structures.

ERGM generates random graphs capturing various structural properties based on a probability distribution defined by sufficient statistics. We use the ERGM implementation in NetworkX to generate random DAGs with a specified expected degree, iteratively adding edges that maintain acyclicity.

The configuration model [47] generates random graphs with a specified degree sequence. We use two variations of the configuration model in our experiments:

1. In the first variation (conf), the degree sequence is set to a constant value equal to the expected degree for all nodes, resulting in graphs with a homogeneous degree distribution.

2. In the second variation (conf2), we use a variable degree sequence sampled from a Poisson distribution with a mean of $q\frac{d}{d-1}$, where $d$ is the number of nodes and $q$ is the expected degree. This allows for generating graphs with heterogeneous degree distributions while maintaining the expected average degree. We ensure that the sum of the sampled degree sequence is even by resampling until this condition is met.

The geometric random graph model [51] generates graphs based on the spatial proximity of nodes in a metric space. Nodes are randomly placed in a unit square, and edges are formed between pairs of nodes if their Euclidean distance is below a specified threshold radius. We calculate the radius using the formula for the radius $r = \sqrt{\frac{q}{\pi(d-1)}}$, where $d$ is the number of nodes and $q$ is the expected degree. This choice of radius ensures that the expected degree is approximately achieved in the resulting graph.

To focus on the impact of graph distributions, we keep the expected degree fixed for evaluation. Comparing results across different graph generations, we find no significant difference in performance, suggesting robustness to variations in graph structure.

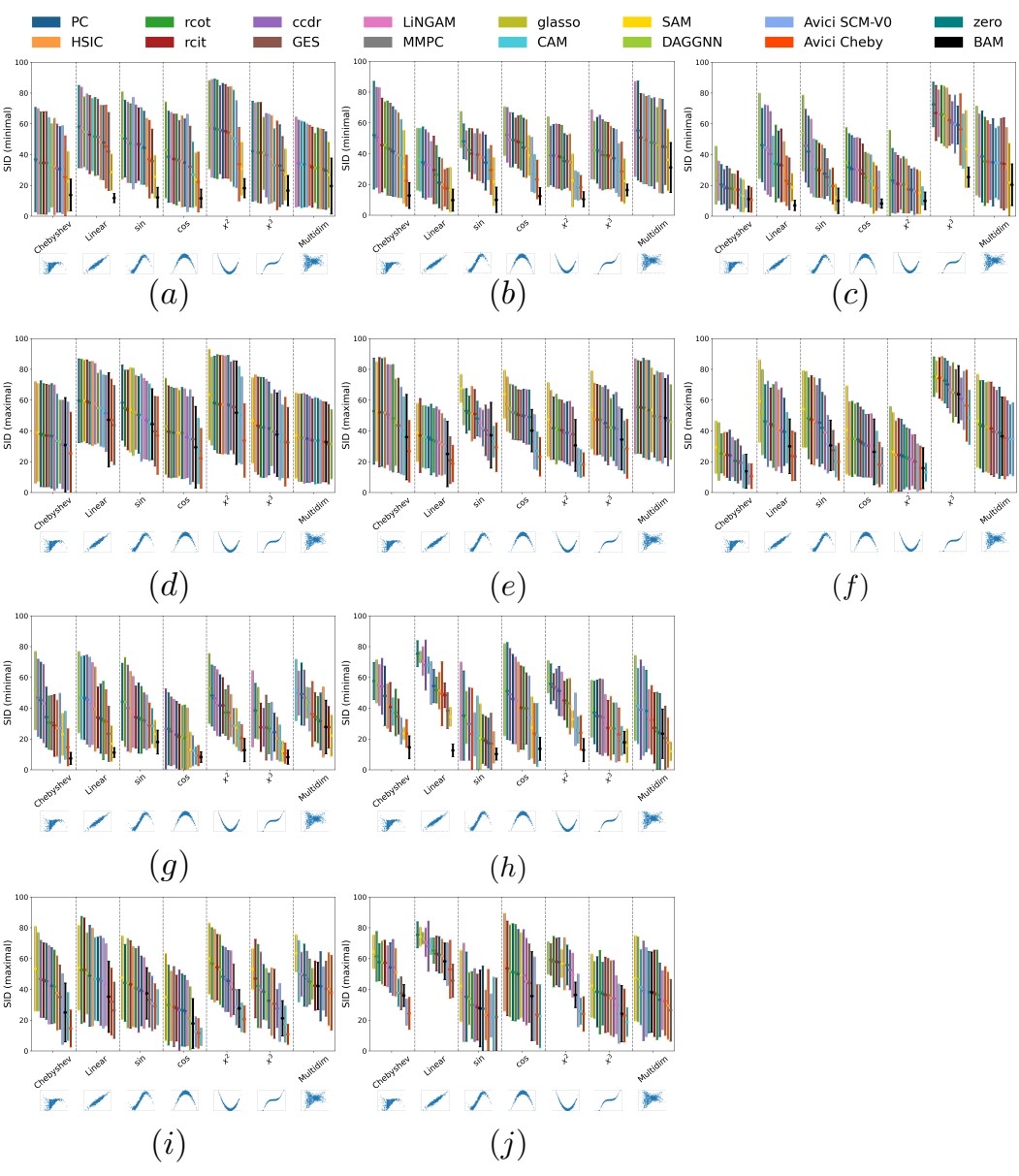

Figure 12: Structural Intervention Distance (SID) results for CPDAG estimation, arranged by SID values from highest (left) to lowest (right) for dimension $d = 10$. Panels (a), (b), (c), (g), and (h) present the minimal SID values, while panels (d), (e), (f), (i), and (j) display the maximal SID values within their respective Markov equivalence classes. The panels correspond to different sample sizes: $M = 50$ for (a) and (d); $M = 100$ for (b) and (e); $M = 200$ for (c) and (f); $M = 500$ for (g) and (i); and $M = 1000$ for (h) and (j).

## G.7 Time comparison

Figure 13 depicts the average runtimes per evaluation step, accompanied by their corresponding standard deviations. We compared BAM with various unsupervised methods, noting that the evaluation times for other supervised approaches, such as Avici, are comparable to those observed for BAM. Specifically, the results are presented in the form of mean $\pm$ one standard deviation. The x-axis enumerates various sample sizes, denoted as $M$, while both mean and standard deviation were computed based on 5 independent inference tests for each configuration with a fixed sample size $M$ and graph dimension $d$.

These empirical observations substantiate the computational efficiency of supervised approaches in the inference phase.

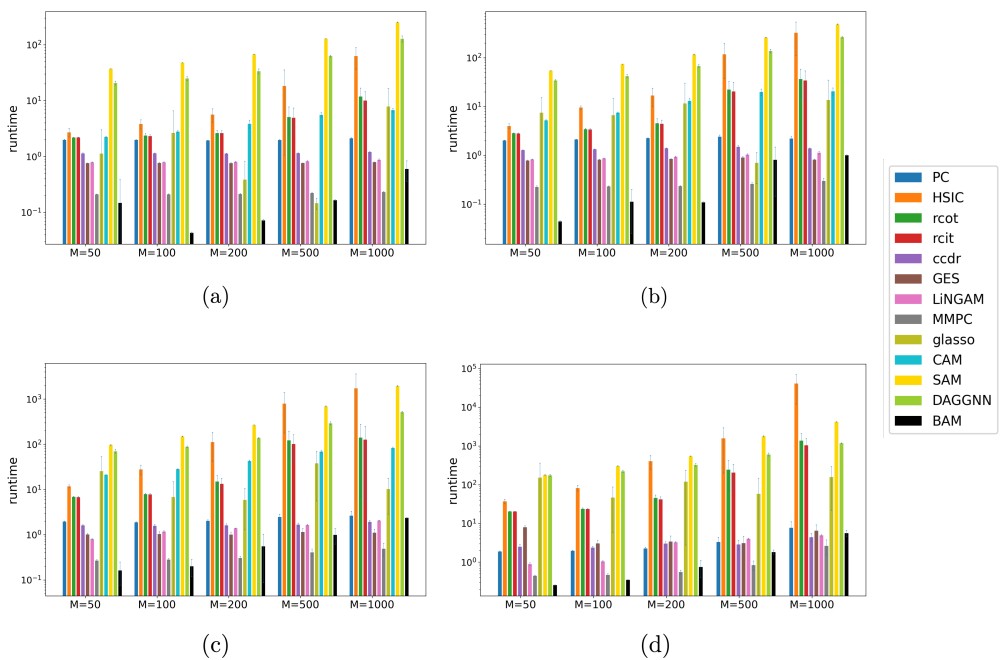

Figure 13: Algorithm runtime for (a) $d = 10$, (b) $d = 20$, (c) $d = 50$, (d) $d = 100$ in seconds per $M \times d$ data matrix inference.

# H Computational complexity analysis

## H.1 Memory complexity

The overall memory complexity of our proposed method is $O(CMd + Cd^2 + Md^2 + M^2d + C^2)$, where $M$ is the number of datapoints, $d$ is the data dimension, and $C$ is the number of channels. This complexity arises from the following components: attention-between-attributes $O(CMd + Md^2)$, attention-between-datapoints $O(CMd + M^2d)$, covariance matrix calculation $O(C(Md + d^2))$, bilinear attention $O(Cd^2)$, and matrix logarithm $O(Cd^2)$. These complexities are derived from the matrix shapes used in Figure 3. Additionally, $C \times C$ weight matrices are used, which require $O(C^2)$ memory.

## H.2 Runtime complexity

The overall runtime complexity of our approach is $O(C^2Md + CMd^2 + CM^2d + C^2d^2 + Cd^3)$. We present a detailed breakdown of the single components:

We first consider the computational complexity of the tensor multiplication defined in (1): a tensor $\boldsymbol{A} \in \mathbb{R}^{M \times d \times C}$ is multiplied by a matrix $\boldsymbol{B} \in \mathbb{R}^{C \times C}$. This operation involves multiplying each of the $M$ slices $\in \mathbb{R}^{d \times C}$ of $\boldsymbol{A}$ by $\boldsymbol{B}$. The multiplication of a $d \times C$ matrix by a $C \times C$ matrix has a time complexity of $O(C^2d)$. Since this multiplication is performed in parallel over the M axis, the total complexity of this tensor multiplication is $O(C^2Md)$. Similarly, the parallel computation of the attention matrix $\boldsymbol{K} \odot \boldsymbol{Q}$ for attention-between-attributes (defined in (4), matrix $\boldsymbol{A}$ in Figure 3 left) consists of $M$ parallel matrix multiplications of $d \times C$ with $C \times d$ matrices, resulting in a complexity of $O(CMd^2)$. Multiplying the attention matrix with the values in (5) is an M-parallel computation of a $M \times d \times d$ tensor with a $M \times d \times C$ tensor, having complexity $O(CMd^2)$. Thus, the overall complexity of attention-between-attributes is $O(C^2Md + CMd^2)$. By switching axes, attention-between-datapoints has a time complexity of $O(C^2Md + CM^2d)$.

Calculating $C$ covariance matrices has a complexity of $O(CMd(\min M, d))$. In a BAM layer, multiplying the $d \times d \times C$ tensor with a $C \times C$ weight matrix has a complexity of $O(C^2d^2)$. The bilinear operation in (2) (for calculating the attention matrix and the output of the BAM layer) is defined as $C$-parallel computation of two $d \times d$ matrix multiplications, resulting in a complexity of $O(Cd^3)$. Computing the custom softmax can be expressed as $d \times d$ matrix multiplications in $C$ channels, so its complexity is $O(Cd^3)$. Calculating the matrix logarithm is equivalent to computing an eigendecomposition, which has a complexity of $O(Cd^3)$ when performed over $C$ channels [20]. In summary, the overall time complexity of our approach is $O(C^2Md + CMd^2 + CM^2d + C^2d^2 + Cd^3)$.

## H.3 Computational efficiency discussion

While our method demonstrates strong performance across various settings, the computational resources required for optimal training (approximately 82 GiB of GPU memory) may present limitations in resource-constrained environments. This is primarily due to the attention mechanisms in our architecture, which exhibit quadratic complexity with respect to sequence length.

For datasets with large sample sizes $M$, the attention-between-datapoints layer becomes the primary computational bottleneck, with memory complexity $O(M^2d)$ and runtime complexity $O(CM^2d)$. To address this limitation and enhance scalability, we propose a local attention strategy: dividing the sample axis $M = ms$ into $s$ subsamples of size $m$ and applying attention within these segments. This modification reduces the runtime complexity to $O(Cm^2ds) = O(\frac{CM^2d}{s})$ and memory complexity to $O(dm^2s)$. By maintaining fixed subsample size $m$, the approach achieves linear scalability in $M$ through linear increments in $s$. Similar principles can be applied to both the attention-between-datapoints and bilinear attention mechanisms.

Furthermore, as demonstrated in Section G.7, our trained model offers significant practical advantages through fast inference times compared to traditional unsupervised approaches, which require iterative optimization for each new dataset.

# I Interpretation

## I.1 Shape-agnostic architecture and the role of attention layers

When employing a shape-agnostic architecture for matrices $\in \mathbb{R}^{M \times d}$, it is crucial to ensure that all elements within the $M \times d$ matrix can interact and influence one another. Consider a scenario where one axis of the matrix is expanded to the shape $\mathbb{R}^{M \times d \times 1}$, followed by dense layers with $1 \times C$ and $C \times C$ weights. In this configuration, the dense layers carry out element-wise operations on the $M \times d$ elements, processing them in isolation from each other. This is because each hidden representation is essentially a linear combination of $C$ matrices of shape $M \times d$ prior to the application of an element-wise activation function.

This limitation is addressed by incorporating attention layers into the architecture. These layers adaptively compute non-trainable $M \times M$ and $d \times d$ attention matrices based on trainable $C \times C$ weights. This approach allows for a permutation- and shape-agnostic architecture, as the same set of trainable weights can be employed for any $M \times d$ matrix, while still enabling the matrix entries to influence each other. In this way, the attention mechanism becomes an essential component of our model. Although we also experimented with Network-in-Network methods [38], we found that the attention mechanism offers a more stable, efficient, and straightforward computation of non-trainable $M \times M$ and $d \times d$ attention matrices, using only trainable $C \times C$ weights.

Our proposed bilinear attention mechanism is, to our knowledge, the first SPD layer to enable shape-agnostic computations. It uses trainable $C \times C$ weights to calculate non-trainable $d \times d$ attention matrices, allowing for adaptive weighting across different $d \times d$ SPD matrix sizes. This flexibility makes it a unique and essential component of our architecture. Additionally, this construction ensures the desired permutation invariance among the variables. Our approach essentially learns matrix operations that should be applicable to any input matrix with arbitrary $M \times d$ input.

## I.2 Attention scores in the BAM layer

Consider the setting as in Figure 3 (right) and the computation of the output $\boldsymbol{H}$ by $\boldsymbol{A} \otimes \boldsymbol{S}$, where $\boldsymbol{A} \in \mathcal{S}_{\succeq}^{d \times d \times C}$ are the attention scores and $\boldsymbol{S} \in \mathcal{S}_{\succeq}^{d \times d \times C}$ are the input matrices into the BAM layer. Since $\boldsymbol{A} \otimes \boldsymbol{S}$ is processed parallel across the channels, we consider for simplicity the output of a single channel here and assume $\boldsymbol{A} \in \mathcal{S}_{\succeq}^{d \times d}$ and $\boldsymbol{S} \in \mathcal{S}_{\succeq}^{d \times d}$ to be quadratic, positive definite $d \times d$ matrices.

While traditional self-attention computes scores to assess the importance of one data point to another, our bilinear attention mechanism extends this by exploring the interdependence of variable pairs. Specifically, for an output pair $(i, j)$, its associated output value is determined not merely by a direct scalar relationship but by the bilinear form: $\sum_{k,l} A_{i,k} S_{k,l} A_{l,j}$. Thus, instead of a singular focus on the relation "How does $j$ affect $i$?", quantified in the score matrix $\boldsymbol{A}$ in classical attention, the score matrix in bilinear attention shows the interaction strengths of pair sets $\{(i, k) \mid k = 1, \ldots, d\}$ and $\{(l, j) \mid l = 1, \ldots, d\}$. The "receptive field" adopts a cross-form within the scores $\boldsymbol{A}$ instead of being $A_{i,j}$ only, in the sense that relevant scores for the output at position $(i, j)$ are not limited to $A_{i,j}$ but $\{\boldsymbol{A}_{i,\cdot} \cup \boldsymbol{A}_{\cdot,j}\}$.

## I.3 Keys and queries

Continuing with the single-channel assumption due to parallel channel processing, consider the quadratic form $\mathcal{S}^{d \times d} \times \mathcal{S}^{d \times d} \mapsto \mathcal{S}^{d \times d}$, $(\boldsymbol{K}, \boldsymbol{Q}) \mapsto \boldsymbol{K}^T \boldsymbol{Q} \boldsymbol{K}$ of the key-query interaction. The $(i, j)$-th entry of $\boldsymbol{K}^T \boldsymbol{Q} \boldsymbol{K}$ is $\boldsymbol{K}_i^T \boldsymbol{Q} \boldsymbol{K}_j$ for the columns $\boldsymbol{K}_1, \ldots, \boldsymbol{K}_d$ of $\boldsymbol{K}$, which are often referred to as keys. Using the eigendecomposition of $\boldsymbol{Q} = \boldsymbol{U}^T \boldsymbol{D} \boldsymbol{U}$ one obtains for the $(i, j)$-th entry the bilinear form $(\boldsymbol{U} \boldsymbol{K}_i)^T \boldsymbol{D} (\boldsymbol{U} \boldsymbol{K}_j)$. Note that $\boldsymbol{U} \boldsymbol{K}_i$ is a similarity measure between $\boldsymbol{K}$ and $\boldsymbol{Q}$ analogous to standard attention. So, for bilinear attention, similarity scores are calculated between the keys $\boldsymbol{K}$ and the eigenvectors of the queries $\boldsymbol{Q}$. Afterwards, the $\boldsymbol{D}^{\frac{1}{2}}$-weighted bilinear-form $(\boldsymbol{D}^{\frac{1}{2}} \boldsymbol{U} \boldsymbol{K}_i)^T (\boldsymbol{D}^{\frac{1}{2}} \boldsymbol{U} \boldsymbol{K}_j)$ is used to create covariance matrices by combining the similarity scores between $\boldsymbol{U}$ and $\boldsymbol{K}$. Hence, in bilinear attention, the similarity scores are functions of both the $i$-th and $j$-th keys as well as all queries. This is consistent with the attention-score behavior, where the

interaction strengths of all pair sets $\{(i, k) \mid k = 1, \ldots, d\}$ and $\{(l, j) \mid l = 1, \ldots, d\}$ collectively influence the output.

This is in contrast to standard attention, which uses the untransformed dot product $(\boldsymbol{k}, \boldsymbol{q}) \in \mathbb{R}^d \times \mathbb{R}^d \mapsto \boldsymbol{k}^T \boldsymbol{q} \in \mathbb{R}$ for the columns of key and query matrices $\boldsymbol{K}, \boldsymbol{Q}$.

## J    Limitations

The proposed model effectively captures smooth dependence relations using Chebyshev polynomials. While this approach demonstrates excellent performance across various types of dependencies, it may face challenges when dealing with data structures that significantly deviate from the generated synthetic data, particularly those with rapidly increasing higher-order derivatives. However, adapting the synthetic data generation process to accommodate these structures is straightforward and can mitigate this limitation.

The Log-Eig layer performs efficiently within a moderate dimensional range, but it may encounter computational constraints when scaling to higher dimensions. Training our model with parameters similar to those used in this study requires substantial memory resources; in our experiments, approximately 80 GB of GPU memory was necessary.

The attention mechanism, although effective, can be computationally expensive for high-dimensional inputs (both in terms of $M$ and $d$). Local attention can alleviate this issue, but it may introduce its own set of challenges. Despite these limitations, the model's architecture allows for easy extensions to address specific applications. For instance, incorporating separate embedding layers for observational and interventional data could enable the model to leverage interventional data effectively.

However, it is important to note that the high computational cost is primarily associated with the training phase, and inference is much faster. While our experiments utilized substantial computational resources (approximately 80 GB of GPU memory), the model architecture allows for effective scaling to more modest computational environments. By reducing key parameters such as the maximum number of samples ($\overline{M}$), maximum data dimension ($\overline{d}$), and number of channels ($C$), training can be successfully conducted on standard computers with suitable GPU support, though potentially with reduced performance in large dataset regimes.

An end-to-end approach for CPDAG estimation could potentially offer further benefits. However, the current model loses directional information during the covariance computation, making an end-to-end approach unfeasible with the existing architecture. A possible extension to facilitate directional inference could involve using two separate embeddings for each variable, one for the variable as a parent and another for the variable as a child, effectively doubling the dimensionality to $2d$.

Another limitation of our approach is the difficulty in theoretically proving the identifiability of causal effects using our neural network method. Identifiability of a causal effect, when having access to the data distribution via a neural network, relies on the fundamental approximation theorem of neural networks, which guarantees that a single-hidden-layer neural network can approximate any continuous function on a compact subset of the input space, given sufficient hidden units [27], such that any mapping $X \mapsto \mathcal{G}$ from the data distribution to graph structures can be learned. However, the theorem might not be directly applicable to the observational attention layers and the bilinear attention layers employed in our approach. Furthermore, for identifiability to be assured, the test data must be sufficiently close to the training data for out-of-sample approximation, which adds another layer of complexity to the problem.

