# OpenReview forum: "Graph Structure Inference with BAM: Neural Dependency Processing via Bilinear Attention"
_NeurIPS.cc/2024/Conference — NeurIPS 2024 poster_

### Official Review · Reviewer_wDfW · 2024-07-11

**Soundness:** 3
**Presentation:** 3
**Contribution:** 3
**Rating:** 6
**Confidence:** 2

**Summary:**

The paper presents a novel neural network model designed for supervised graph structure inference. The core contribution is the introduction of a Bilinear Attention Mechanism (BAM) which processes dependency information at the level of covariance matrices of transformed data. This approach respects the geometry of the manifold of symmetric positive definite matrices, leading to robust and accurate inference of both undirected graphs and partially directed acyclic graphs (CPDAGs). The model is trained on variably shaped and coupled simulated input data and requires only a single forward pass for inference. Empirical evaluations demonstrate the model's robustness in detecting a wide range of dependencies, outperforming several state-of-the-art methods.

**Strengths:**

- The introduction of the Bilinear Attention Mechanism (BAM) is novel and aims to address key challenges in graph structure inference.
- The empirical evaluations are comprehensive, demonstrating the robustness and effectiveness of the proposed method.
- The paper is well-written. The figures depicted contribute to clearly demonstrating the pipeline proposed.
- The method provides a significant improvement over existing approaches.

**Weaknesses:**

- The method's complexity, particularly in terms of the bilinear attention mechanism and its application, may limit its accessibility to a broader audience.
- The training process, requiring extensive computational resources, may pose a challenge for researchers with limited access to high-performance computing facilities.

**Questions:**

1. How does the performance of the model scale with larger and more complex graph structures?
2. Please summarize the research motivation and major contributions of this work.
3. Can the authors provide more details on the computational requirements and potential optimizations for the training process?
4. Considering the high computational cost and significant memory resources required during the training phase (e.g., approximately 80 GB of GPU memory), what strategies or architectural modifications do you plan to implement to mitigate these computational demands without compromising the model's performance in the future?

**Limitations:**

The authors have adequately discussed the limitations and potential negative societal impact of their work. However, more detailed discussions on the scalability and computational requirements could be beneficial. Constructive suggestions for improvement include exploring optimizations for the training process and providing additional real-world case studies to validate the method's applicability.

---

> ### Author Rebuttal · Authors · 2024-08-06
>
> Dear Reviewer wDfW,
>
> Thank you for the insightful and constructive feedback. We sincerely appreciate your recognition of the novelty and effectiveness of our Bilinear Attention Mechanism (BAM) for graph structure inference.
>
> **Research motivation:** We agree that the paper will benefit from a clearer explanation of the research motivation, particularly regarding why we introduce the BAM layer in the SPD space for graph structure inference. We have addressed this in the overall author rebuttal.
>
> **Major contributions:** Our primary contribution lies in the introduction of the novel Bilinear Attention Mechanism (BAM) layer. Additionally, we propose the innovative application of geometric deep learning methods, inspired by the computer vision field, in the SPD space for supervised causal learning. Moreover, we introduce a novel learning task that respects identifiability and distinguishes between moralized and skeleton edges, enabling more precise causal structure inference. In the final version of our paper, we will enhance our presentation to ensure that these essential contributions are more readily accessible to the reader.
>
> **Computational complexity and resource requirements**
> We acknowledge the potential limitations of our method in terms of its computational complexity and the requirement for extensive computational resources during training, which is a common challenge in deep learning, particularly when compared to the vast computational architectures employed by large companies in fields such as computer vision and GPT models.
>
> Nevertheless, we would like to highlight that our approach can be effectively scaled down to accommodate more modest computational resources. By reducing model and training generator parameters such as the maximum number of samples ($\overline{M}=500$), maximum data dimension ($\overline{d}=50$), and number of channels ($C=50$), we have successfully conducted training on a standard MacBook equipped with an M1 chip and 16 GB memory. We utilized Apple Metal to facilitate GPU-like operations on the M1 chip. This approach is also feasible on computers equipped with a suitable NVIDIA graphics card, offering an alternative for efficient training without the need for a dedicated computing server.
> Although this scaled-down version may not achieve the same level of performance as the high-resource setup for large dataset regimes with large $M$ and $d$ values, it showcases the flexibility and adaptability of our approach to various computational constraints.
>
> **Memory complexity:**
> The overall memory complexity of our proposed method is $O(CMd + Cd^2 + Md^2 + M^2d+C^2)$, where $M$ is the number of datapoints, $d$ is the data dimension, and $C$ is the number of channels. This complexity arises from the following components: attention-between-attributes $O(CMd+Md^2)$, attention-between-datapoints $O(CMd+M^2 d)$, covariance matrix calculation $O(C(Md+d^2))$, bilinear attention $O(Cd^2)$, and matrix logarithm $O(Cd^2)$. These complexities are derived from the matrix shapes used in Figure 3 of the paper. Additionally, $C\times C$ weight matrices are used, which require $O(C^2)$ memory. We thank the reviewer for drawing our attention to the incorrect memory complexity given in line 311, which will be corrected in the final version of the paper.
>
>
> **Computational time complexity:**
> We thank the reviewer for the suggestion to also add an analysis of the computational complexity. The runtime complexity of our approach is $O(C^2 M d+CMd^2+CM^2 d +C^2 d^2 + Cd^3)$. We present a detailed breakdown of the single components:
>
> The complexity of attention-between-attributes is $O(C^2 Md+CMd^2)$. By switching axes, attention-between-datapoints has a time complexity of $O(C^2 Md+CM^2d)$.
>
> In a BAM layer, multiplying the $d\times d\times C$ tensor with a $C\times C$ weight matrix has a complexity of $O(C^2 d^2)$. The bilinear operation in line 174 (for calculating the attention matrix and the output of the BAM layer) is defined as $C$-parallel computation of two $d\times d$ matrix multiplications, resulting in a complexity of $O(Cd^3)$. Computing the custom softmax can be expressed as $d\times d$ matrix multiplications in $C$ channels, so its complexity is $O(Cd^3)$. Calculating the matrix logarithm is equivalent to computing an eigendecomposition, which has a complexity of $O(Cd^3)$ when performed over $C$ channels.
>
>
> **Handling larger datasets:**
> We acknowledge the importance of reducing computational complexity for large, high-dimensional datasets. We propose using local attention instead of global attention to address those cases.
>
> Consider a dataset with a large sample size $M$. In such cases, the attention-between-datapoints layer becomes a bottleneck due to its $O(M^2 d)$ memory complexity and $O(CM^2 d)$ computational complexity. To alleviate this, we propose dividing the sample axis $M = ms$ randomly into $s$ subsamples, each containing $m$ samples. Applying local attention to these smaller segments reduces the runtime complexity to $O(Cm^2 ds)=O(\frac{CM^2d}{s})$ and the memory complexity to $O(dm^2s)$ for calculating the attention matrix between samples. By maintaining a fixed subsample size $m$, our approach achieves linear scalability with respect to $M$ by incrementing linearly in $s$. This technique can be similarly applied to the attention-between-datapoints and our bilinear attention methods, enhancing overall efficiency.
>
> In the revised manuscript, we will incorporate the computational complexity analysis and provide a detailed discussion on efficiency in the appendix. Furthermore, we will address the limitations related to computational complexity, including the potential of local attention as a mitigating approach.
>
> Best,
>
> Authors of Submission15606

---

### Official Review · Reviewer_2mp1 · 2024-07-13

**Soundness:** 3
**Presentation:** 2
**Contribution:** 3
**Rating:** 6
**Confidence:** 2

**Summary:**

This paper studies the problem of graph structure inference using a neural network with bilinear attention mechanism.

**Strengths:**

The proposed framework is novel and can operate in Euclidean and positive semi-definite matrix spaces. Attention mechanism is utilized in an innovative manner to reveal interdependencies. Experiments demonstrate that the proposed framework achieves competent and robust performance.

**Weaknesses:**

1. Could you clarify whether BAM has any theoretical guarantees on the recovery of correct structure? In the absence of such guarantees, it is not clear how BAM might perform in settings beyond those explored in the paper.

2. The advantages of BAM over the state-of-the-art approaches could be explained better from a conceptual perspective.

**Questions:**

See Weaknesses.

**Limitations:**

Limitations have been adequately discussed.

---

> ### Author Rebuttal · Authors · 2024-08-06
>
> Dear Reviewer 2mp1,
>
>
> Thank you for recognizing the novelty and innovation of our framework. We appreciate the constructive and insightful feedback, on clarifying the advantages of our BAM layer in the SPD space and the theoretical guarantees of our approach.
>
> **Significance and advantages of SPD layers:**
> We agree that the paper will benefit from a clearer justification of the integration of SPD layers and the Bilinear Attention Mechanism (BAM) in our neural network architecture, highlighting their advantages. We have addressed this in the overall author rebuttal.
>
> **Theoretical guarantees:**
> Thank you for raising the important question of whether our BAM framework provides theoretical guarantees on the recovery of the correct graph structure. We acknowledge that the current paper does not include formal proofs of identifiability or consistency. We have addressed the challenges in theoretically proving identifiability and consistency in the Limitations section. The universal approximation theorem for neural networks provides some assurance that, in principle, a sufficiently expressive neural network can learn any continuous mapping from the data distribution to graph structures. However, several assumptions need to be fulfilled to ensure convergence to such a solution.
>
> Moreover, the theorem pertains to performance on the training data, and generalization to out-of-sample data is a different challenge. Rigorously extending these results to our specific architecture with observational attention and bilinear attention layers is non-trivial.
>
> That said, we believe our extensive empirical evaluation provides compelling evidence that BAM is able to reliably recover meaningful graph structures in practice. Across a range of settings, BAM achieved strong performance competitive with or exceeding state-of-the-art baselines. This situation, where empirical performance outpaces theoretical understanding, has been a common phenomenon in the deep learning field. Many deep learning approaches have achieved state-of-the-art results in various domains despite limited theoretical foundations. While we absolutely agree that developing a rigorous theoretical understanding is crucial and the lack thereof is a clear limitation, we also believe that research in deep learning and its applications to new domains should not entirely be avoided due to the absence of a complete theoretical framework.
>
> We will incorporate a more detailed discussion on the advantages of SPD layers and the BAM architecture, as well as an extended analysis of the limitations regarding theoretical guarantees, in the revised manuscript and appendix, following your suggestions.
>
> Best,
>
> Authors of Submission15606

---

> > ### Comment · Reviewer_2mp1 · 2024-08-07
> >
> > I thank the authors for the rebuttal. After considering it and other reviews, I have raised my score.

---

> > > ### Author Response · Authors · 2024-08-07
> > >
> > > Thank you for considering our rebuttal and for raising your score.

---

### Official Review · Reviewer_rbW5 · 2024-07-14

**Soundness:** 3
**Presentation:** 4
**Contribution:** 3
**Rating:** 6
**Confidence:** 4

**Summary:**

This paper proposes to use supervised causal learning approach to learn causal structure. It takes a dataset as input, and outputs a moral graph. The moral graph is an undirected graph with two types of edges: skeleton edge and moralized edge.

In technical space:
architecture: The approach adopts alternating node-wise and sample-wise attention + covariance in embedded space + bilinear attention via SPD net
training data: ER sampling to generate graphs, Chebyshev polynomial functions as basis to generate mechanism and with Gaussian distribution for noise terms.
learning task: The main task is a three-class edge classification to output a moral graph.

experiment shows good performance by comparing with other SOTA competitors, for both the moral graph prediction and CPDAG prediction.

**Strengths:**

1. Supervised causal learning is an interesting area, and it has potentially more impact to the causal discovery community.

2. The learning task respects the identifiability, and the main task is to predict a moral graph, which is new to the best of my knowledge.

3. A novel NN architecture is interesting and could make potential impact to the community.

**Weaknesses:**

1.	Superiority of model architecture: What is the significance and necessity of using semi-definite (SPD) matrix space? How to "enhancing graph structure inference" should be explained in more detail. Compared with only using alternating attention, what advantages does BAM have and what scenarios are more suitable? The theoretical discussion and experiments in the current paper are not enough to explain the above problems.

2.	The necessity of introducing Moralized edges: Overall I like the idea of representing the undirected graph by the moral graph, which is new and sound. However, given that the paper also has a second prediction model to do orientation from moral graph to yield the final CPDAG, so, what are the advantages compared to learning skeleton and v-structure separately? So, I would like to see more discussion and some empirical evidence to demostrate the advantages of choosing moralized graph instead of skeleton as the key learning task.

3.	The motivation for choosing Chebyshev as the mechanism of training data: For example, in "E.1 Chebyshev polynomials for training", this Chebyshev polynomial form does not seem to be able to describe complex multivariate coupling effects. In addition, there should be corresponding experiments for the hyperparameter setting of Chebyshev to prove that these training effects are robust or sensitive to this hyperparameter.

4.	In the experimental part, a large amount of synthetic data is used to test performance, but the functions (mechanisms) of these synthetic data are relatively simple. More complex or more representative mechanisms should be used for testing to enhance persuasiveness, such as MLP, Random Fourier features, etc.

5.	In addition, the experimental part is all tested on synthetic data, and should be tested on real benchmarks, such as Sachs, etc.

**Questions:**

See my comments on the weakness part above.

---

> ### Author Rebuttal · Authors · 2024-08-06
>
> Dear Reviewer rbW5,
>
> Thank you for the insightful and constructive feedback. We sincerely appreciate your recognition of the potential impact of our approach.
>
> **Significance of SPD layers:**
> We agree that the paper will benefit from a clearer justification of the integration of SPD layers and the Bilinear Attention Mechanism (BAM) in our neural network architecture. We have addressed this in the overall author rebuttal.
>
> **Evidence for the effectiveness of moralized edge estimation:**
> Thank you for acknowledging our contribution in distinguishing between skeleton and moralized edges in the undirected edge estimation task. We agree that empirical evaluation would further demonstrate its effectiveness. To address this, we ran additional experiments where we maintained the undirected edge estimation network but modified the second step for estimating immoralities. We trained another neural network to test each possible immorality in the graph, not just those identified by the first neural network, using the same hyperparameters for training as before. The results, shown in Table 1 of the rebuttal PDF (at the bottom of the author rebuttal), indicate that our approach of only testing immoralities detected by the first neural network consistently outperformed the case where all possible moralizations were tested.
>
> We observed that the neural network testing all possible moralizations had a strong bias towards estimating no moralization, likely because this occurs more frequently in sparse graphs used for training. While this bias could potentially be addressed using a weighted loss function, it highlights the efficiency of our approach in avoiding these issues. Information about conditional dependencies might be easier to detect "globally" (i.e., having information about the whole graph, not only the Markov blanket estimated beforehand), akin to the Gaussian case, where this information is contained in the precision matrix.
>
> **Multidimensional couplings:**
> We appreciate the reviewer's suggestion to test our approach on more complex and representative mechanisms. We agree that this would enhance the persuasiveness of our experimental results. To address this point, we conducted additional experiments, as shown in Figure 1 of the rebuttal PDF. In these experiments, we tested our approach for detecting Random Fourier feature dependencies.
>
> For the undirected case, shown in (A, B, C) of Figure 1, our algorithm outperforms the baselines for all sample sizes except the two highest, $M=500, 1000$, where our method still produces very good results. We believe that for this sample size, our approach might overfit the training data, becoming more confident in not estimating an edge when it behaves differently from what was seen during training.
>
> Regarding the directed estimation results shown in (D, E, F), our algorithm still exhibits reasonable performance, although it is not the best when compared to the baselines. This highlights opportunities for improvement for multidimensional dependencies, which we will discuss in the limitations section.
> Additionally we have already tested our approach on MLP dependencies, as shown in Appendix G.2 of our paper, where we demonstrate competitive performance of our approach.
>
> **Rationale behind using Chebyshev polynomials:**
>
> Chebyshev polynomials exhibit factorially decreasing coefficients when approximating functions with bounded derivatives of any order. By sampling the coefficients from a uniform distribution with factorial decay, we can generate coupling functions that are characterized by being smooth and well-behaved with a large degree of variation. This property ensures that the generated coupling functions are diverse and well-behaved, without being dominated by high-frequency oscillations or other unstable behavior.
>
> The orthogonality of Chebyshev polynomials is a desirable property for our application, as it allows us to easily evaluate the magnitude of individual effects and ensures that the effects do not interfere with each other in an unstable manner. This is particularly important to control the magnitude of the causal effects to ensure that an edge in the adjacency matrix corresponds to a meaningful, controllable effect in the generated data. Training with disrupted training data, where an edge is to be predicted without being predictable is causing problems for learning.
>
> In contrast, using e.g. a randomly initialized MLP as coupling may not provide the same level of control and interpretability. While the universal approximation theorem suggests that MLPs can represent a wide range of functions, the expressive power of a randomly initialized MLP is limited. The weights are typically drawn from simple distributions like Gaussian or uniform, which can lead to similar patterns in the network's behavior across different initializations. As a result, the simulated relationships may be opaque and limited expressive.
>
> Our experiments demonstrate that Chebyshev polynomials can approximate a wide range of simple and complex dependencies, including random MLP and RFF dependencies. However, we acknowledge our algorithm's limitations in detecting multidimensional dependencies, particularly in the RFF dependency case for directed edge estimation.
>
> **Evaluation on real benchmarks:**
> We acknowledge the suggestion to test on real-world benchmark datasets. While we recognize the importance of testing on real data, due to time constraints and the scope of this submission, we focused on synthetic data to validate our method.
>
> For real-world data, the true graph structure is often uncertain, and there are many random effects, making principled evaluation challenging. Synthetic data allows for a more controlled evaluation of detection accuracy. Nevertheless, we agree that real-world data are crucial and will be the next step in our future studies.
>
> We will incorporate all points discussed above in the final version of our paper.
>
> Best,
>
> Authors of Submission15606

---

### Official Review · Reviewer_wqVr · 2024-07-31

**Soundness:** 3
**Presentation:** 3
**Contribution:** 3
**Rating:** 7
**Confidence:** 3

**Summary:**

This paper proposes a novel neural network model for supervised graph structure inference. The model aims to learn the mapping between observational data and their underlying dependence structure using a bilinear attention mechanism (BAM). The BAM operates on the level of covariance matrices of transformed data and respects the geometry of the manifold of symmetric positive definite (SPD) matrices. The paper demonstrates the robustness of the proposed method in detecting dependencies and inferring graph structures.

**Strengths:**

1. The introduction of the bilinear attention mechanism for processing dependency information in the SPD space is novel and well-motivated. This approach leverages the geometric properties of SPD matrices, offering a fresh perspective on graph structure inference.

2. The paper provides extensive experimental results, showing that the proposed method outperforms state-of-the-art algorithms in various scenarios.

**Weaknesses:**

Although the paper claims that the proposed method is computationally efficient compared to some unsupervised approaches, the actual computational cost, especially during training, is unclear.

**Questions:**

1. Provide a more detailed analysis of the theoretical computational complexity and practical scaling ability of the proposed method.

2. Discuss potential strategies for optimizing the computational efficiency and handling larger datasets effectively.

---

> ### Author Rebuttal · Authors · 2024-08-05
>
> Dear Reviewer wqVr,
>
> Thank you for the insightful and constructive feedback. We greatly appreciate your recognition of the potential of our approach.
>
> **Efficiency:**
> We acknowledge that the level of computational resources required for our method (approximately 82 GiB, as stated in lines 309-312) may be a limiting factor in cases where such resources are not readily available. This is indeed an advantage of traditional causal learning algorithms. Deep learning models, particularly attention-based architectures, which exhibit quadratic complexity with respect to sequence length, require significant computational resources to achieve optimal performance. However, when coupled with large computational resources, deep learning has demonstrated superior performance across various domains, such as natural language processing (e.g., GPT-4) and computer vision (e.g., state-of-the-art object detection and segmentation models).
>
>
> **Memory complexity:**
> The overall memory complexity of our proposed method is $O(CMd + Cd^2 + Md^2 + M^2d+C^2)$, where $M$ is the number of datapoints, $d$ is the data dimension, and $C$ is the number of channels. This complexity arises from the following components: attention-between-attributes $O(CMd+Md^2)$, attention-between-datapoints $O(CMd+M^2 d)$, covariance matrix calculation $O(C(Md+d^2))$, bilinear attention $O(Cd^2)$, and matrix logarithm $O(Cd^2)$. These complexities are derived from the matrix shapes used in Figure 3 of the paper. Additionally, $C\times C$ weight matrices are used, which require $O(C^2)$ memory. We thank the reviewer for drawing our attention to the incorrect memory complexity given in line 311, which will be corrected in the final version of the paper.
>
>
> **Computational time complexity:**
> We thank the reviewer for the suggestion to also add an analysis of the computational complexity. The runtime complexity of our approach is $O(C^2 M d+CMd^2+CM^2 d +C^2 d^2 + Cd^3)$. We present a detailed breakdown of the single components:
>
> We first consider the computational complexity of the tensor multiplication defined in lines 127-128: a tensor $\boldsymbol{A}\in\mathbb{R}^{M\times d \times C}$ is multiplied by a matrix $\boldsymbol{B}\in \mathbb{R}^{C\times C}$. This operation involves multiplying each of the $M$ slices $\in\mathbb{R}^{d\times C}$ of $\boldsymbol{A}$ by $\boldsymbol{B}$. The multiplication of a $d\times C$ matrix by a $C\times C$ matrix has a time complexity of $O(C^2 d)$. Since this multiplication is performed in parallel over the M axis, the total complexity of this tensor multiplication is $O(C^2 M d)$.
> Similarly, the parallel computation of the attention matrix $\boldsymbol{K}\odot\boldsymbol{Q}$ for attention-between-attributes (line 498, matrix $\boldsymbol{A}$ in Figure 3 left) consists of $M$ parallel matrix multiplications of $d\times C$ with $C\times d$ matrices, resulting in a complexity of $O(CMd^2)$. Multiplying the attention matrix with the values (line 503) is an M-parallel computation of a $M\times d\times d$ tensor with a $M\times d \times C$ tensor, having complexity $O(CMd^2)$. Thus, the overall complexity of attention-between-attributes is $O(C^2 Md+CMd^2)$. By switching axes, attention-between-datapoints has a time complexity of $O(C^2 Md+CM^2d)$.
>
> Calculating $C$ covariance matrices has a complexity of $O(CMd(\min{M,d}))$. In a BAM layer, multiplying the $d\times d\times C$ tensor with a $C\times C$ weight matrix has a complexity of $O(C^2 d^2)$. The bilinear operation in line 174 (for calculating the attention matrix and the output of the BAM layer) is defined as $C$-parallel computation of two $d\times d$ matrix multiplications, resulting in a complexity of $O(Cd^3)$. Computing the custom softmax can be expressed as $d\times d$ matrix multiplications in $C$ channels, so its complexity is $O(Cd^3)$. Calculating the matrix logarithm is equivalent to computing an eigendecomposition, which has a complexity of $O(Cd^3)$ when performed over $C$ channels.
> In summary, the overall time complexity of our approach is $O(C^2 M d+CMd^2+CM^2 d +C^2 d^2 + Cd^3)$.
>
> **Handling larger datasets:**
> We acknowledge the importance of reducing computational complexity for large, high-dimensional datasets. We propose using local attention instead of global attention to address those cases.
>
> Consider a dataset with a large sample size $M$. In such cases, the attention-between-datapoints layer becomes a bottleneck due to its $O(M^2 d)$ memory complexity and $O(CM^2 d)$ computational complexity. To alleviate this, we propose dividing the sample axis $M = ms$ randomly into $s$ subsamples, each containing $m$ samples. Applying local attention to these smaller segments reduces the runtime complexity to $O(Cm^2 ds)=O(\frac{CM^2d}{s})$ and the memory complexity to $O(dm^2s)$ for calculating the attention matrix between samples. By maintaining a fixed subsample size $m$, our approach achieves linear scalability with respect to $M$ by incrementing linearly in $s$. This technique can be similarly applied to the attention-between-datapoints and our bilinear attention methods, enhancing overall efficiency.
>
>
> In the revised manuscript, we will incorporate the computational complexity analysis and provide a detailed discussion on efficiency in the appendix. Furthermore, we will address the limitations related to computational complexity, including the potential of local attention as a mitigating approach.
>
> Best,
>
> Authors of Submission15606

---

> > ### Comment · Reviewer_wqVr · 2024-08-13
> >
> > Thank you for your detailed response. It effectively addresses my questions. I will maintain my acceptance rating.

---

> > > ### Author Response · Authors · 2024-08-13
> > >
> > > Thank you for your response and your positive assessment.

---

### Author Rebuttal · Authors · 2024-08-06

We appreciate the time and effort the reviewers have invested in evaluating our work. We are grateful for your constructive feedback and insightful suggestions, which have helped us identify areas for improvement.

We have added a PDF at the end of the author rebuttal, which includes a comparison of the identification of moralizations and evaluations on data using Random Fourier feature dependencies.

In response to the reviewers' requests for further elaboration on the motivation and rationale behind the advantages of our approach, we provide a more detailed explanation of the advantages of our approach.

**Motivation for Optimizing in the SPD Space for Graph Estimation: GLASSO optimization:**

We acknowledge the importance of clearly justifying the integration of SPD-layers in the neural network architecture.

The use of SPD layers is inspired by the success of traditional GLASSO-based methods for undirected graph estimation, such as the influential works of Banerjee et al. (2008), Friedman et al. (2008), and Yuan and Lin (2007). These methods optimize a penalized likelihood over the SPD space using techniques like block coordinate descent, ensuring that the estimate remains positive definite at each iteration. These approaches have proven to be efficient for learning graph structures when the input SPD matrix encodes sufficient information, particularly in the Gaussian case where the covariance matrix is a sufficient statistic for the distribution (neglecting a shift by the mean). The main reason optimization is performed solely in the SPD space is that inverse covariance matrices naturally represent graphs. Zero entries indicate missing edges and non-zero entries correspond to the strength of conditional dependencies.  This characteristic makes them a powerful tool for estimating well-conditioned and interpretable structures within the SPD space, particularly when combined with $l_1$ regularization to promote sparsity.

However, for general SEMs with non-linear dependencies and non-Gaussian error terms, pure SPD models cannot capture the full information of the data. Neural networks, on the other hand, excel at learning and decoding complex non-linear relationships. The observational attention layers in our model are trained to nonlinearly transform the data into a batch of $C$ covariance matrices, learning an end-to-end observational transformation that filters the information useful for graph prediction and provides sufficient information for efficiently solving the undirected graph estimation task.

Thus, our BAM-network learns data-driven matrix transformations akin to those used in solving the GLASSO optimization problem. Our network implicitly encodes the solution to a parametric optimization problem. For some objective function $F$ (e.g. penalized negative log-likelihood for GLASSO), we optimize $\min_{\Sigma} F(\Sigma, \mathcal{D})$ with respect to $\Sigma$, where the solution is parametric in the data $\mathcal{D}$, or any sufficient statistics thereof. The solution describes a mapping $\mathcal{D} \rightarrow \hat{\Sigma}$, with $\hat{\Sigma} = \arg\min_{\Sigma} F(\Sigma, \mathcal{D})$ being the space of all optimization solutions for different data $\mathcal{D}$. Hence, instead of solving the SPD GLASSO optimization for each new dataset $\mathcal{D}$ to find the corresponding $\hat{\Sigma}$, the network learns the entire mapping $\hat{\Sigma}(\mathcal{D})$, requiring only a single cheap forward pass through a neural network for evaluation on a specific $\mathcal{D}$. Additionally, a transformation in the observational data space to handle non-Gaussian data is learned at the beginning. It is worth noting that the operations for solving GLASSO problems by a traditional algorithm can be applied to input data of any dimension $d$ and any sample size $M$. Our network also achieves this by applying only attention-based, shape-invariant operations.

Another advantage of our end-to-end learning approach over traditional likelihood-based methods is that it does not require a sensitive sparsity hyperparameter.

**Advantages of Optimizing in the SPD Space for Graph Estimation: Enhanced expressiveness and natural representation:**

In traditional supervised causal discovery, the adjacency matrix values are directly calculated from a learned $M \times d\times C$ transformed observational data matrices. This requires decoding the $d \times d$ pairwise relationships within each of the $M$ samples that far, that probabilities can be obtained after a few simple operations. Before the output is calculated, the $M$ axis is max-pooled out (Lorch et al., 2022) or cut out (Ke et al., 2022), resulting in a $C\times d$ data representation in both cases. However, extracting $d(d-1)/2$ distinct pairwise interactions from a $C \times d$ representation by one operation on the $C$-axis lacks expressiveness especially when the data dimension $d$ varies. Also, using Max-Pooling for the sample axis might be problematic for varying sample sizes $M$, because the magnitude of the output values scales with $M$ alone.

In contrast, using a $d \times d$ representation naturally accommodates the varying dimensionality and allows for a more direct and efficient extraction of the pairwise dependencies. We argue that using $C \times d$ representations for inferring $d \times d$ pairwise interactions is structurally misaligned, leading to inefficiencies compared to directly operating on $d \times d$ matrices.

**References**:

Banerjee et al. (2008). Model selection through sparse maximum likelihood estimation for multivariate Gaussian or binary data. JMLR

Friedman et al. (2008). Sparse inverse covariance estimation with the graphical lasso. Biostatistics

Ke et al. (2022). Learning to Induce Causal Structure. In ICLR.

Lorch et al. (2022). Amortized inference for causal structure learning. In NeurIPS

Yuan \& Lin (2007). Model selection and estimation in the Gaussian graphical model. Biometrika

---

### Decision · Program_Chairs · 2024-09-25

**Decision:**

Accept (poster)

**Comment:**

This paper introduces a new neural network model for supervised graph structure inference, featuring a novel Bilinear Attention Mechanism (BAM) that effectively processes dependency information at the level of covariance matrices of transformed data. The proposed method shows promise for learning causal structures, with experimental results sufficient to support its effectiveness. While there were initial concerns regarding several aspects, including the algorithm's complexity, the authors' rebuttal has adequately addressed many of the reviewers' questions. Given these considerations, I believe the paper has sufficient merit to be accepted for this venue.